# Simultaneous sulfide and methane oxidation by an extremophile

Rob A. Schmitz [1,2], Stijn H. Peeters[1], Sepehr S. Mohammadi[1], Tom Berben[1], Timo van Erven[1], Carmen A. Iosif[1], Theo van Alen[1], Wouter Versantvoort [1], Mike S. M. Jetten [1], Huub J. M. Op den Camp [1] ✉ & Arjan Pol[1]

Hydrogen sulfide ($H_2S$) and methane ($CH_4$) are produced in anoxic environments through sulfate reduction and organic matter decomposition. Both gases diffuse upwards into oxic zones where aerobic methanotrophs mitigate $CH_4$ emissions by oxidizing this potent greenhouse gas. Although methanotrophs in myriad environments encounter toxic $H_2S$, it is virtually unknown how they are affected. Here, through extensive chemostat culturing we show that a single microorganism can oxidize $CH_4$ and $H_2S$ simultaneously at equally high rates. By oxidizing $H_2S$ to elemental sulfur, the thermoacidophilic methanotroph *Methylacidiphilum fumariolicum* SolV alleviates the inhibitory effects of $H_2S$ on methanotrophy. Strain SolV adapts to increasing $H_2S$ by expressing a sulfide-insensitive $ba_3$-type terminal oxidase and grows as chemolithoautotroph using $H_2S$ as sole energy source. Genomic surveys revealed putative sulfide-oxidizing enzymes in numerous methanotrophs, suggesting that $H_2S$ oxidation is much more widespread in methanotrophs than previously assumed, enabling them to connect carbon and sulfur cycles in novel ways.

Hydrogen sulfide ($H_2S$) is the most reduced form of sulfur (S) and a potent energy and sulfur source, toxicant, and signaling molecule[1–3]. It is a weak acid that easily diffuses through membranes and inhibits various processes such as aerobic respiration by binding to cytochrome *c* oxidases. In addition, other metabolic processes that use copper- and iron-containing enzymes are severely inhibited by $H_2S$[1,4–6]. Hence, microorganisms living in sulfide-rich environments require adequate mechanisms to detoxify $H_2S$[7,8]. In a myriad of environments, such as wetlands, marine sediments, soil, wastewater treatment plants, lakes, paddy fields, landfills, and acidic geothermal environments, $H_2S$ is produced through sulfate ($SO_4^{2-}$) reduction, mineralization of organic matter, and thermochemistry[8–18].

Upon depletion of sulfate, organic matter is ultimately converted to methane ($CH_4$) in oxygen-depleted ecosystems[9,12,13,19–21]. When both $H_2S$ and $CH_4$ diffuse into the overlaying oxic zones, $CH_4$ can be utilized as an energy source by aerobic methane-oxidizing bacteria, which are assumed to mitigate most emissions of this potent greenhouse gas[22]. Despite this effective methane biofilter, 548 to 736 Tg of $CH_4$ is annually released into the atmosphere from various natural and anthropogenic sources[23,24]. Aerobic methanotrophs are part of various bacterial classes and families, including the ubiquitous Alpha- and Gammaproteobacteria[16,25,26], Actinobacteria[27] and the extremophilic *Methylacidiphilaceae* of the phylum Verrucomicrobia[28–31]. The latter are acidophilic bacteria that share a low pH optimum (2.0 – 3.5) and live between 35 and 60 °C[26,31,32]. All known verrucomicrobial methanotrophs have been isolated from geothermal habitats such as fumaroles and mudpots, from which large amounts of mostly thermogenic $CH_4$ and $H_2S$ are emitted[16,28,33–35]. Geothermal environments are typically characterized by high $H_2S$ emissions and thus the verrucomicrobial methanotrophs isolated from these ecosystems are preeminent examples to study how methanotrophs are affected by $H_2S$.

[1]Department of Microbiology, Radboud Institute for Biological and Environmental Sciences, Radboud University, Heyendaalseweg 135, 6525AJ Nijmegen, The Netherlands. [2]Present address: Institute of Biogeochemistry and Pollutant Dynamics, Department of Environmental Systems Science, ETH Zurich, 8092 Zurich, Switzerland. ✉e-mail: h.opdencamp@science.ru.nl

It is becoming increasingly clear that methanotrophs are metabolically versatile and able to use environmentally relevant energy sources such as $H_2$, propane, ethane, acetate, acetone, 2-propanol, and acetol[16,36–38]. The ability to utilize various energy sources is highly beneficial in environments with heavily fluctuating gas emissions. Recently, it was demonstrated that pure cultures of the verrucomicrobial methanotroph *Methylacidiphilum fumariolicum* SolV can consume methanethiol ($CH_3SH$), with the concomitant sub-stoichiometric formation of $H_2S$, indicating that strain SolV partially metabolized toxic $H_2S$[39]. Hereafter, an elegant study demonstrated that also proteobacterial methanotrophs can oxidize $H_2S$[40]. The authors isolated the versatile alphaproteobacterium 'Methylovirgula thiovorans' strain HY1 from a South Korean peatland that could grow on thiosulfate ($S_2O_3^{2-}$), tetrathionate ($S_4O_6^{2-}$), elemental sulfur ($S^0$), and a range of carbon compounds. However, strain HY1 cells grown on $CH_4$ as sole energy source were not able to oxidize $H_2S$, and $H_2S$ oxidation was only initiated and observed in cells grown in the presence of thiosulfate. In addition, growth on $H_2S$ was not studied. Considering recent observations, it is paramount to investigate whether microbes exist that can oxidize the environmentally relevant gases $CH_4$ and $H_2S$ simultaneously, how methanotrophs cope with $H_2S$ and whether such methanotrophs can conserve energy and produce biomass using $H_2S$ as an energy source.

Here, through extensive chemostat cultivation, we show for the first time that a microorganism can oxidize $CH_4$ and $H_2S$ simultaneously. *M. fumariolicum* SolV is inhibited by the presence of elevated $H_2S$ concentrations but $H_2S$ is rapidly oxidized to elemental sulfur ($S^0$) as a detoxification mechanism to alleviate the inhibitory effect of $H_2S$ on $CH_4$ oxidation. Strain SolV adapts to $H_2S$ with the upregulation of a Type III sulfide:quinone oxidoreductase (SQR) and an $H_2S$-insensitive $ba_3$-type cytochrome $c$ oxidase, creating an electron transfer pathway from $H_2S$ to $O_2$. Additionally, strain SolV incorporates $^{13}CO_2$ using $H_2S$ as sole energy source. We propose that the $H_2S$ oxidation capacity of verrucomicrobial methanotrophs is essential to thrive in sulfur-rich acidic geothermal ecosystems. In addition, we found SQR in a plethora of proteobacterial methanotrophs of various environments. Considering $CH_4$ and $H_2S$ coexist in a myriad of oxygen-limited ecosystems, $H_2S$ oxidation could be a trait present among many aerobic methanotrophs.

## Results

### Simultaneous $H_2S$ and $CH_4$ oxidation, and chemolithoautotrophic growth on $H_2S$

The detection of genes encoding putative sulfide:quinone oxidoreductases (SQRs) in the genomes of verrucomicrobial methanotrophs prompted us to investigate whether methanotrophs can oxidize and adapt to $H_2S$[16]. Accordingly, a continuous culture of the thermoacidophilic aerobic methanotroph *Methylacidiphilum fumariolicum* SolV (running as chemostat at a dilution rate (D) of 0.016 h$^{-1}$) was maintained with $CH_4$ as energy source and $CO_2$ as carbon source (non-adapted cells; Fig. 1a), up to a load of 39 µmol $CH_4$ min$^{-1}$ · g DW$^{-1}$ (Table 1). For comparison, a distinct continuous cultivation system was designed (with identical $CH_4$ load) to adapt cells to increasing loads of $H_2S$ (Supplementary Fig. 1). The cells growing in this chemostat simultaneously oxidized $H_2S$ and $CH_4$ (sulfide-adapted cells; Fig. 1b), up to concurrent loads of 42 µmol $H_2S$ · min$^{-1}$ · g DW$^{-1}$ and of 38 µmol $CH_4$ · min$^{-1}$ · g DW$^{-1}$ (Table 1), while the $H_2S$ concentration in the gas outlet remained below 2 nmol · L$^{-1}$. Steady state continuous cultures of non-adapted and sulfide-adapted cells could be maintained for many generations (Fig. 1a, b). Accumulation of elemental sulfur ($S^0$) over weeks of growth was evident, as increasing amounts of a yellow precipitate (irregular microscopic particles) developed and attached to the metal parts and walls of the chemostat (Supplementary Fig. 2). After a few weeks of operation with $H_2S$ it was identified as being over 99% pure sulfur and the amount could account for at least 80% of the sulfide converted over this period. Through microscopy, only minute amounts of sulfur particles in the liquid could be observed as opposed to bacterial cells. Both non-adapted and sulfide-adapted cultures were operated under low dissolved $O_2$ concentrations (1% air saturation) to minimize chemical sulfide oxidation. The low $O_2$ concentrations also resulted in expression of hydrogenase activity as observed previously[41]. Control incubations in membrane-inlet mass spectrometry (MIMS) experiments without cells showed negligible oxidation of sulfide at micromolar range concentrations.

Verrucomicrobial methanotrophs possess the Calvin-Benson-Bassham cycle for $CO_2$ fixation[42], raising the question whether they can grow as chemolithoautotroph on $CO_2$ with $H_2S$ as energy source. Accordingly, a fed-batch reactor was inoculated with a diluted culture ($OD_{600} = 0.05$) of the dual $H_2S$-$CH_4$ chemostat and $H_2S$ and $^{13}CO_2$ were supplemented as the only energy and carbon source, while the $CH_4$

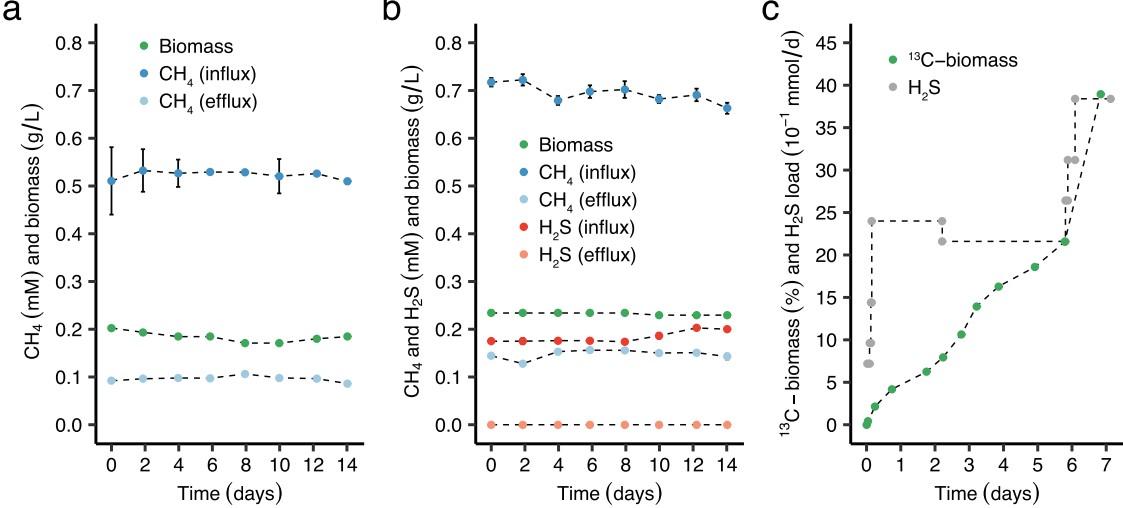

**Fig. 1 | Growth of *M. fumariolicum* SolV at high loads of $CH_4$ only, $H_2S$ and $CH_4$, or $H_2S$ only. a** Continuous culture oxidizing methane. **b** Continuous culture simultaneously oxidizing high concentrations of $CH_4$ and $H_2S$. **c** Fed-batch culture showing increase in $^{13}C$-biomass with $H_2S$ as sole energy source. Data are presented as mean ± standard deviations ($n$ = 3 technical replicates). Source data are provided as a Source Data file.

**Table 1 | Comparison of conversion and respiration rates of *M. fumariolicum* SolV cells from the CH$_4$ chemostat (non-adapted cells) and the dual H$_2$S-CH$_4$ chemostat (sulfide-adapted cells)**

| | non-adapted cells | sulfide-adapted cells |
|---|---|---|
| Conversion rates in the chemostat [a] | | |
| CH$_4$ conversion | 39 | 38 |
| H$_2$S conversion | – | 42 |
| Max. H$_2$S conversion (at <0.15 µM H$_2$S and 1.7 µM O$_2$) | – | 156 |
| Maximum conversion rates in the MIMS chamber [b] | | |
| CH$_4$ conversion | 200 ± 11 | 133 ± 9 |
| H$_2$ conversion | 78 – 104 | 60 – 82 |
| H$_2$S conversion (at 5–30 µM H$_2$S and <10 µM O$_2$) | 22 ± 4 | 120 ± 13 |
| H$_2$S conversion (at 5–30 µM H$_2$S and 60–80 µM O$_2$) | – | 132 – 154 |
| Maximum respiration rates in the MIMS chamber [b] | | |
| CH$_4$ respiration [c] | 302 ± 9 | 211 ± 11 |
| CH$_3$OH respiration | 311 ± 22 | 211 ± 13 |
| H$_2$ respiration | 29 – 36 | 18 – 31 |
| H$_2$S respiration (at 40–80 µM H$_2$S and <10 µM O$_2$) | 10 ± 1 | 53 ± 4 |
| H$_2$S respiration (at 30–80 µM H$_2$S and 70–90 µM O$_2$) | 14 ± 1 | 77 ± 4 |

[a]Measured using GC and calculated from the differences between the gas inlet and gas outlet of the chemostat.
[b]Measured through membrane inlet mass-spectrometry (MIMS) and a fiber-optic oxygen sensor spot.
[c]This rate includes the theoretical 1 mol O$_2$ needed to activate 1 mol CH$_4$.
All rates are in µmol · min$^{-1}$ · g DW$^{-1}$. All CH$_4$, CH$_3$OH and H$_2$ conversion and respiration rates measured in the MIMS chamber were determined in the absence of H$_2$S. Respiration refers to O$_2$ consumption in response to addition of the listed substrates.

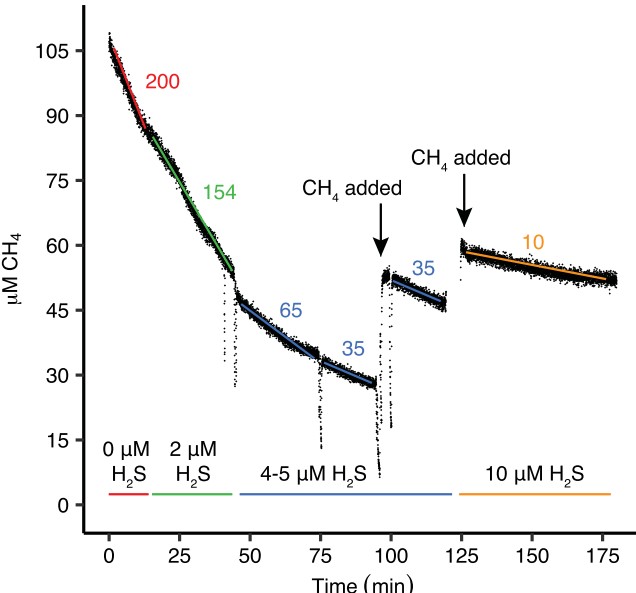

**Fig. 2 | Inhibition of CH$_4$ consumption by non-adapted *Methylacidiphilum fumariolicum* SolV cells in the presence of H$_2$S.** H$_2$S was kept at various stable concentrations (indicated at the bottom) by pulse-wise additions of H$_2$S to the MIMS chamber. Numbers indicate CH$_4$ consumption rates in µmol CH$_4$ · min$^{-1}$ · g DW$^{-1}$. At 170 min the MIMS chamber has become anoxic, resulting in cessation of CH$_4$ consumption. Source data are provided as a Source Data file.

supply was disconnected. Over time, the biomass of *M. fumariolicum* SolV cells became enriched in carbon-13 by incorporating $^{13}$CO$_2$ into biomass (Fig. 1c). When the H$_2$S load was increased, the percentage of $^{13}$C-biomass increased accordingly. Growth was evident, as an increase in $^{13}$C-biomass was accompanied with an increase in dry weight (Supplementary Fig. 3). By quantifying H$_2$S in the gas inlet and outlet of the reactor, H$_2$S conversion efficiencies of ~98–100% were determined throughout the whole incubation period.

### H$_2$S inhibition, oxidation, and adaptation to H$_2$S

H$_2$S consumption rates and inhibitory effects of H$_2$S on *M. fumariolicum* SolV cells were measured inside a liquid-filled chamber connected to a membrane-inlet mass spectrometer (MIMS), which allows for the real-time and concurrent quantification of multiple gases, while O$_2$ was measured by a sensor spot. A maximum CH$_4$ conversion rate of non-adapted cells of 200 ± 11 µmol CH$_4$ · min$^{-1}$ · g DW$^{-1}$ was measured with a concomitant O$_2$ consumption rate of 302 ± 9 µmol O$_2$ · min$^{-1}$ · g DW$^{-1}$ (Table 1). In comparison, for the sulfide-adapted cells a maximum CH$_4$ conversion rate and concomitant O$_2$ consumption rate of 33 and 30% lower was measured, respectively. Similarly, the maximum methanol respiration rates of sulfide-adapted cells were 32% lower than measured for the non-adapted cells (Table 1). Taking the 1 mol O$_2$ required for the activation of 1 mol CH$_4$ into account, the maximum CH$_4$ respiration rates of the non-adapted and sulfide-adapted cells were about 3-fold lower compared to maximum methanol respiration rates (Table 1), indicating the conversion of methane to methanol as the rate limiting step. In addition, presumably due to the low dO$_2$ concentration in the continuous cultures, the non-adapted and sulfide-adapted cells expressed a high hydrogenase activity (Table 1), with a measured H$_2$:O$_2$ consumption ratio of ~1:0.35 as expected[32,42]. As was

the case for CH$_4$ and methanol respiration, the maximum H$_2$ respiration rates of the sulfide-adapted cells were lower than those of the non-adapted cells (Table 1). Hence, the gain in increased H$_2$S oxidation capacity in sulfide-adapted cells comes at the expense of the CH$_4$, methanol and H$_2$ conversion capacities.

Sulfide-adapted cells in the chemostat oxidized H$_2$S to low, non-inhibitory concentrations (Fig. 1b), which is necessary since the CH$_4$ oxidation capacity of non-adapted cells (as well as sulfide-adapted cells) was affected by an H$_2$S concentration as low as 1 µM. CH$_4$ oxidation was inhibited by about 25%, 70–85% and 95% in the presence of 2 µM, 4-5 µM and 10 µM H$_2$S, respectively (Fig. 2). Inhibition of CH$_4$ conversion appeared reversible, as when H$_2$S was consumed or flushed out of short-term incubations, CH$_4$ conversion and CO$_2$ production resumed immediately at their previous rates. After longer periods (2 h) of inhibition by 10–20 µM H$_2$S, CH$_4$ conversion rates were 25–35% lower. Whether these lower rates were the result of inhibition of pMMO or other parts of the respiratory chain as well could not be concluded as methanol (CH$_3$OH) conversion was impaired as well after such long H$_2$S exposures.

High initial O$_2$ consumption rates were measured when only H$_2$S was administered to non-adapted cells in the MIMS chamber. Interestingly, these rates immediately and rapidly decreased ~15-fold within a few minutes to stable rates of 10 ± 1 µmol O$_2$ · min$^{-1}$ · g DW$^{-1}$ (at 40–80 µM H$_2$S and <10 µM O$_2$). This rapid decrease in respiration rate indicated the presence of sulfide-sensitive terminal oxidases (SSTOs) that were quickly inactivated after the addition of H$_2$S and at least one type of sulfide-insensitive terminal oxidase (SITO) responsible for the remaining low respiration rate[43]. The maximum reaction rate of the SITO (10 ± 1 µmol O$_2$ · min$^{-1}$ · g DW$^{-1}$) is limited, as it constitutes only 3% of the maximum respiration rate of these non-adapted cells on methanol (Table 1). At 10-fold higher O$_2$ concentrations (70–90 µM O$_2$ and 30–80 µM H$_2$S), the remaining respiration rate increased ~40% (Table 1), suggesting that O$_2$ is competing with H$_2$S for the active site of the SSTOs, thereby alleviating H$_2$S inhibition. SITO activity was cyanide sensitive as 95% of the respiration rate was inhibited at 1 mM potassium cyanide. The sulfide-adapted cells oxidized H$_2$S with maximum O$_2$

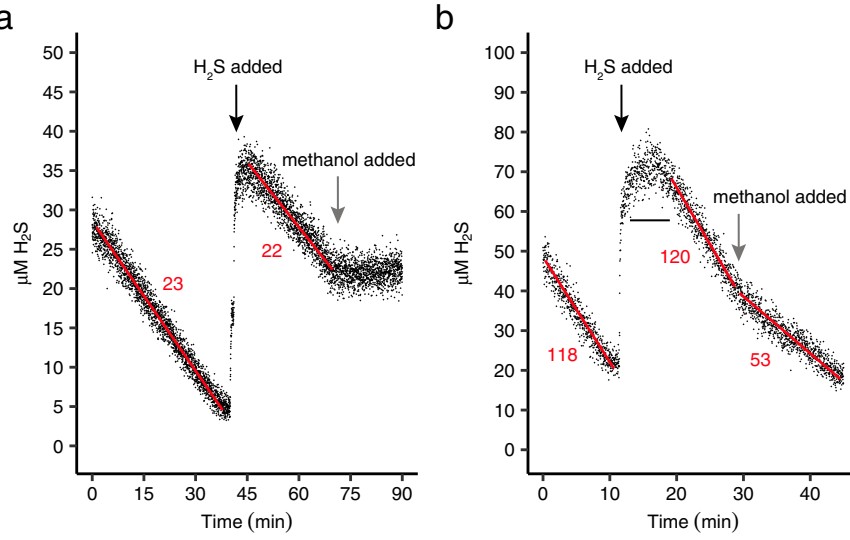

**Fig. 3 | Inhibition of $H_2S$ consumption by *Methylacidiphilum fumariolicum* SolV cells in the presence of methanol. a** Cessation of $H_2S$ consumption by non-adapted cells after the addition of methanol (final concentration 0.4 mM). **b** Inhibition of $H_2S$ consumption by sulfide-adapted cells after the addition of methanol (final concentration 5 mM). Numbers indicate consumption rates in µmol $H_2S \cdot min^{-1} \cdot g \ DW^{-1}$. The black horizontal line indicates a brief moment of anoxia to demonstrate $H_2S$ oxidation is dependent on $O_2$. Source data are provided as a Source Data file.

consumption rates of $53 \pm 4$ µmol $O_2 \cdot min^{-1} \cdot g \ DW^{-1}$ (Table 1). As at 40–80 µM $H_2S$ the SSTOs were assumed to be completely inhibited, these values represent the rates of the SITO, which are more than five times higher compared to the non-adapted cells (Table 1). $H_2S$ is primarily converted to elemental sulfur ($S^0$), as a $H_2S:O_2$ stoichiometry of 1:0.48 ($\pm 0.005$; $n = 3$) was determined after simultaneous quantification of $H_2S$ and $O_2$ consumption, together with the visible production of $S^0$ (Supplementary Fig. 7b).

Maximum conversion rates of $H_2S$ at non-inhibiting, low (sub-micromolar) concentrations in the MIMS chamber were difficult to perform due to its rapid consumption that resulted in a variable inhibition. Alternatively, the maximum $H_2S$ conversion rates were determined in the dual $H_2S$-$CH_4$ chemostat by gradually increasing the sulfide load to 156 µmol $H_2S \cdot min^{-1} \cdot g \ DW^{-1}$ over the course of a day while monitoring the outlet concentration (Table 1). The latter increased from 2 to 25 nmol $\cdot L^{-1}$ and therefore remained below a liquid concentration of 40 nM, which was considered not to affect pMMO (as measured through MIMS incubations). Nevertheless, $CH_4$ conversion did decrease about 40% but remained stable for days. When in a similar way the chemostat was given only $H_2S$ while $CH_4$ was disconnected, the same maximum $H_2S$ conversion rate of 156 µmol $H_2S \cdot min^{-1} \cdot g \ DW^{-1}$ was measured. As respiration is not the limiting factor in this setup (Table 1), this rate is considered the maximum $H_2S$ conversion rate, which is 1.5 times higher than the SITO activity can account for in these sulfide-adapted cells and made possible by the SSTOs that were only partially inhibited at these low sulfide concentrations. Similar rates were measured for sulfide-adapted cells in the MIMS chamber in the presence of 60–80 µM $O_2$ (Table 1). Hence, at low $H_2S$ and/or high $O_2$ concentrations, the cells demonstrate the highest sulfide conversion rates, as the SSTOs are only partially inhibited. Noticeably, the above measured maximum $H_2S$ conversion rate in the MIMS chamber exceeded that of the maximum $CH_4$ conversion rate of the sulfide-adapted cells (Table 1).

**Oxidation of methanol, $H_2$ and formic acid in the presence of $H_2S$**

Upon addition of methanol during respiration of 20–40 µM $H_2S$ by non-adapted cells in the MIMS chamber, $H_2S$ consumption ceased immediately (Fig. 3a) but the total respiration rate increased by ~40%. In contrast, $H_2S$ oxidation by sulfide-adapted cells (having five times higher SITO activity) continued at 43% of the rate when methanol was

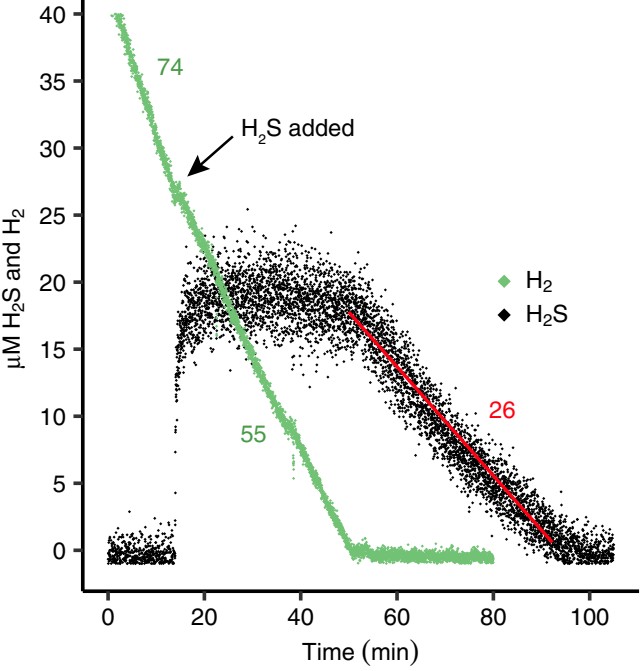

**Fig. 4 | $H_2$ and $H_2S$ consumption dynamics in non-adapted *Methylacidiphilum fumariolicum* SolV cells.** Green numbers indicate $H_2$ consumption rates in µmol $\cdot min^{-1} \cdot g \ DW^{-1}$ before and after $H_2S$ addition, respectively. The red number and line indicate $H_2S$ consumption rate in µmol $\cdot min^{-1} \cdot g \ DW^{-1}$ after depletion of $H_2$. Source data are provided as a Source Data file.

added (Fig. 3b), while the total respiration rate increased by ~25% (Supplementary Fig. 4). Hence, methanol and $H_2S$ were respired simultaneously and seem to compete for the same terminal oxidase. When sulfide (30 µM) was added to sulfide-adapted cells during methanol respiration, $O_2$ consumption decreased ~3-fold (Supplementary Fig. 5). The remaining respiration rates (66 µmol $O_2 \cdot min^{-1} \cdot g \ DW^{-1}$) were higher than expected from the maximum (SITO-dependent) $H_2S$ respiration rate ($53 \pm 4$ µmol $O_2 \cdot min^{-1} \cdot g \ DW^{-1}$), indicating that at least some methanol was still respired, which was confirmed by

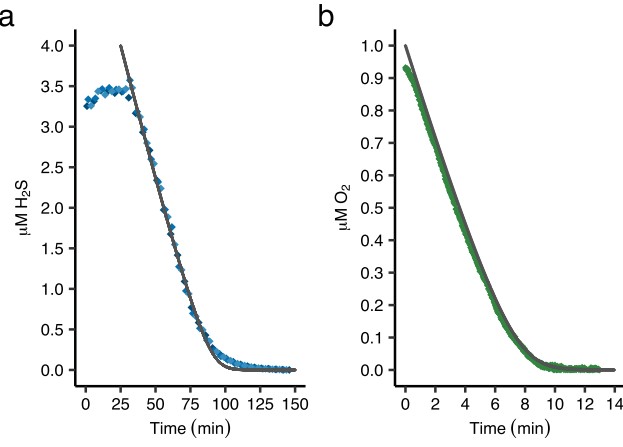

**Fig. 5 | Kinetics of H₂S oxidation by *Methylacidiphilum fumariolicum* SolV cells.** **a** H₂S oxidation measured through gas chromatography. Different blue shaded diamonds represent biological replicates (*n* = 3). The reaction was initiated by addition of cells after 33 min. **b** H₂S respiration measured through a fiber-optic oxygen sensor spot in the MIMS chamber. Black lines indicate Michaelis-Menten curve fitting. The reaction was initiated by addition of cells at 0 min. Source data are provided as a Source Data file.

the fact that the $CO_2$ production rate continued at 20–30% in the presence of sulfide. In contrast, at the same $H_2S$ concentrations, $CH_4$ respiration had ceased almost completely (Fig. 2).

Addition of $H_2S$ to non-adapted cells consuming $H_2$ decreased the $H_2$ consumption rate by 30%, while $H_2S$ consumption only started after complete depletion of $H_2$ (Fig. 4). Furthermore, $H_2$ respiration was up to two times higher than $H_2S$ respiration after all $H_2$ had become depleted. Similarly, $H_2S$ oxidation at 20–40 µM $H_2S$ was reduced by about 80% upon addition of 200 µM formic acid (CHOOH) (Supplementary Fig. 6), while the total respiration rate increased by 15%. The observation that $H_2S$ is only oxidized after $H_2$ (or methanol) has become depleted (Fig. 4) suggests competitive electron transfer pathways to the sulfide-insensitive terminal oxidase (SITO) due to its limited respiration capacity. Interestingly, when in a separate experiment 1.2 mM $H_2S$ was oxidized as sole energy source over ~3 h and $O_2$ additions were stopped, $H_2S$ was produced under anoxic conditions (Supplementary Fig. 7). Conceivably, a sulfide-producing enzyme is being used by strain SolV, reducing the heretofore produced polysulfides and/or elemental sulfur ($S^0$). In contrast, $H_2$ was not consumed in the absence of polysulfides and/or elemental sulfur under anoxic conditions, indicating that these sulfur compounds and not sulfate present in the medium was used as electron acceptor. $H_2S$ production was stimulated up to 13 µmol $H_2S \cdot min^{-1} \cdot g\,DW^{-1}$ when $H_2$ or methanol was present as electron donor.

## Kinetics of H₂S oxidation and respiration
$H_2S$ oxidation kinetics by sulfide-adapted cells were studied using a gas chromatograph, which has a lower detection limit than the MIMS. Starting at 3.5 µM $H_2S$ and 190 µM $O_2$, an almost linear decrease down to 1 µM $H_2S$ was observed with a rate of 167–223 µmol $H_2S \cdot min^{-1} \cdot g\,DW^{-1}$ (Fig. 5a). These rates are slightly higher than the maximum $H_2S$ conversion rates measured in the MIMS chamber at 5–30 µM $H_2S$ and 60–80 µM $O_2$ (Table 1), which could be explained by the higher $O_2$ and low $H_2S$ concentrations present in the incubations used for GC measurements. Because $O_2$ and $H_2S$ compete for the sulfide-sensitive terminal oxidase (SSTO), a low $H_2S$ and high $O_2$ concentration alleviate SSTO inhibition, leading to higher $H_2S$ consumption rates. Michaelis-Menten modeling of $H_2S$ consumption resulted in an apparent affinity constant $K_s$ of 0.32 µM $H_2S$. However, the $H_2S$ traces below 1 µM $H_2S$ did not follow the predicted curve and remained slightly above it

(Fig. 5a). When in the MIMS chamber $O_2$ consumption was followed down to zero at $H_2S$ concentrations of 15–20 µM an apparent affinity constant $K_s$ of $0.14 \pm 0.01$ µM $O_2$ was determined that follows Michaelis-Menten kinetics (Fig. 5b). In the presence of 15 µM $H_2S$, SITO was not inhibited as identical $O_2$ consumption rates were measured after sequential addition of $O_2$ (Supplementary Fig. 8). Assuming only one terminal oxidase type to be active under these conditions and $H_2S$ conversion not being the limiting factor, a $K_s$ of $0.14 \pm 0.01$ µM $O_2$ could be a reliable value for the sulfide-insensitive terminal oxidase.

## Gene regulation in response to H₂S
To assess how *M. fumariolicum* SolV cells adapt to $H_2S$, mRNA from the dual $H_2S$-$CH_4$ chemostat (sulfide-adapted cells) and the $CH_4$ chemostat (non-adapted cells), both in steady state, were extracted and gene expression was quantified (Table 2; Supplementary Fig. 9; Supplementary Data 1). In sulfide-adapted cells, the operon MFUM_v2_0219-21 was upregulated about 1.7-fold. The genes in this operon are annotated as NAD(FAD)-dependent dehydrogenase (MFUM_v2_0219), a protein homologous to the sulfur carrier protein TusA (MFUM_v2_0220) and a putative sulfur carrier protein DsrE2 (MFUM_v2_0221), respectively. A more detailed investigation revealed MFUM_v2_0219 to encode a type III sulfide:quinone oxidoreductase (SQR)[44]. Based on gene comparisons, a second gene (MFUM_v2_0138) might encode an SQR, although this gene was not significantly upregulated in the presence of $H_2S$ and transcribed to a much smaller degree than MFUM_v2_0219 in sulfide-adapted cells (Supplementary Data 1). Two genes (MFUM_v2_0873 and MFUM_v2_1149) were transcribed that might encode sulfur dioxygenases, which could putatively oxidize elemental sulfur to sulfite ($SO_3^{2-}$) (Supplementary Data 1). In addition, the genes MFUM_v2_0942 and MFUM_v2_0943 were upregulated 2-fold and 8-fold (Table 2) and show high similarity to genes encoding the cytochrome *c* protein SorB and sulfite:cytochrome *c* oxidoreductase SorA of *Thiobacillus novellus*, respectively[45]. In sulfide-adapted cells, the putative sulfur dioxygenase (MFUM_v2_0873) was transcribed to a similar degree as SQR (MFUM_v2_0219). However, based on the stoichiometry of 1 $H_2S$: 0.48 $O_2$ ($\pm 0.005$; *n* = 3) quantified in the MIMS chamber, the conversion of elemental sulfur and polysulfides via sulfite to sulfate is thought to have a minor role under the tested conditions. In addition, the oxidation of $H_2S$ was never accompanied by a decrease in pH, which would have been the case if elemental sulfur had been oxidized further to thiosulfate, sulfite or sulfate. The operon MFUM_v2_1257-61 encodes a $ba_3$-type cytochrome *c* oxidase that was upregulated ~5-fold in the presence of $H_2S$, agreeing with the 5-fold higher SITO respiration rate in sulfide-adapted cells. Interestingly, the highest upregulated gene (15-fold) encodes an 89 kDa heptahaem cytochrome *c* protein of unknown function (MFUM_v2_1950), showing highest similarity to genes found in thermophilic sulfide-oxidizers. In the presence of $H_2S$, several genes encoding enzymes involved in the biosynthesis of sulfide for production of sulfur-containing metabolites (e.g., cysteine, methionine and glutathione) were heavily downregulated (Table 2). In addition, the downregulation of genes involved in $CH_4$ oxidation and subsequent electron transfer in the respiratory chain was observed (Table 2). This downregulation is in accordance with the measured decreased maximum methane conversion and respiration rates.

## Phylogeny of putative SQRs in methanotrophs
The observation that *M. fumariolicum* SolV possesses an SQR and the fact that $CH_4$ and $H_2S$ coexist in a large variety of environments prompted us to investigate the presence of SQR in methanotrophs. Indeed, genes encoding putative SQRs are also widespread in proteobacterial methanotrophs of various environments such as lakes, wetlands, rhizosphere, ocean sediments, permafrost soil, paddy fields, wastewater treatment plants, alkaline soda lakes, landfills and groundwater aquifers (Supplementary Fig. 10). SQRs are classified into six different types based on their structure, and differ in their affinity

**Table 2 | Regulation of genes of *M. fumariolicum* SolV cells grown in the dual CH$_4$-H$_2$S chemostat (sulfide-adapted cells) versus the CH$_4$ chemostat (non-adapted cells)**

| ORF | Annotation | Upregulation factor |
|---|---|---|
| Genes involved in the oxidation of sulfur compounds and the respiratory chain | | |
| MFUM_v2_0219 | Sulfide:quinone oxidoreductase (sqr) | 1.8 |
| MFUM_v2_0220 | Putative sulfur carrier protein | 1.7 |
| MFUM_v2_0221 | Peroxiredoxin family protein | 1.7 |
| MFUM_v2_0942 | Cytochrome c family protein | 1.9 |
| MFUM_v2_0943 | Sulfite oxidase or related enzyme | 8.3 |
| MFUM_v2_1257 | Conserved transmembrane protein of unknown function | 4.5 |
| MFUM_v2_1258 | Conserved transmembrane protein of unknown function | 5.4 |
| MFUM_v2_1259 | Cytochrome c oxidase (B(O/a)3-type) chain II (cbaB) | 5.2 |
| MFUM_v2_1260 | Cytochrome c oxidase (B(O/a)3-type) chain I (cbaA) | 4.5 |
| MFUM_v2_1261 | Conserved protein of unknown function | 3.2 |
| MFUM_v2_1950 | Heptahaem-containing protein | 15.7 |
| MFUM_v2_1951 | Putative starvation-inducible outer membrane lipoprotein | 9.7 |

| ORF | Annotation | Downregulation factor |
|---|---|---|
| Genes involved in assimilatory sulfide production | | |
| MFUM_v2_0525 | Sulfate adenylyltransferase subunit 1 | 11.6 |
| MFUM_v2_0526 | Sulfate adenylyltransferase subunit 2 (cysD) | 18.7 |
| MFUM_v2_0527 | Phosphoadenosine phosphosulfate reductase (cysH) | 29.4 |
| MFUM_v2_0528 | Homocitrate synthase 1 (nifV) | 8.6 |
| MFUM_v2_0573 | Polysulfide reductase | 2.2 |
| MFUM_v2_0815 | Sulfite reductase [NADPH] hemoprotein beta-component (cysI) | 33.4 |
| MFUM_v2_2220 | O-acetylserine sulfhydrylase A (cysK) | 3.0 |
| Genes involved in methane oxidation | | |
| MFUM_v2_1464 | PqqA peptide cyclase PqqE | 2.1 |
| MFUM_v2_1604 | Methane monooxygenase subunit alpha (pmoB3) | 3.6 |
| MFUM_v2_1605 | Methane monooxygenase subunit beta (pmoA3) | 3.0 |
| MFUM_v2_1606 | Methane monooxygenase subunit gamma (pmoC3) | 2.9 |
| MFUM_v2_1791 | Methane monooxygenase subunit alpha (pmoB1) | 1.6 |
| MFUM_v2_1792 | Methane monooxygenase subunit beta (pmoA1) | 1.7 |
| MFUM_v2_1793 | Methane monooxygenase subunit gamma (pmoC1) | 1.6 |
| Genes involved in the respiratory chain | | |
| MFUM_v2_2064 | Succinate dehydrogenase flavoprotein subunit | 1.6 |
| MFUM_v2_2065 | Succinate dehydrogenase cytochrome b subunit | 2.5 |
| MFUM_v2_2239 | NADH-quinone oxidoreductase subunit D (nuoD) | 1.6 |
| MFUM_v2_2240 | NADH-quinone oxidoreductase subunit C (nuoC) | 1.6 |
| MFUM_v2_2241 | NADH-quinone oxidoreductase subunit B (nuoB) | 1.6 |
| MFUM_v2_2458 | ATP synthase F1 complex subunit alpha (atpA) | 1.7 |
| MFUM_v2_2459 | ATP synthase F1 complex subunit gamma (atpG) | 1.6 |
| MFUM_v2_2460 | ATP synthase F1 complex subunit beta (atpD) | 1.6 |
| MFUM_v2_1602 | Phosphoenolpyruvate synthetase (ppsA) | 2.7 |

Listed genes have a basemean >4, an upregulation factor or downregulation factor >1.5 and an adjusted *p*-value ≤ 0.05 (all averages of triplicates). A two-sided Wald test was performed by DEseq2 to calculate adjusted *p*-values. *ORF* open reading frame.

for H$_2$S and their physiological role in the cell[44]. Putative SQRs were detected in a large variety of proteobacterial genera in which a pMMO and/or sMMO was present, such as *Crenothrix, Methylobacter, Methylocaldum, Methylocapsa, Methylococcus, Methylocystis, Methylohalobius, Methylomagnum, Methylomarinum, Methylomicrobium, Methylomonas, Methyloprofundus, Methylosinus, Methylospira, Methyloterricola, Methylotetracoccus, Methylotuwimicrobium* and *Methylovulum* (Supplementary Fig. 10). In addition, the recently isolated alphaproteobacterium '*Methylovirgula thiovorans*' strain HY1 encodes a type I SQR[40]. In contrast, verrucomicrobial methanotrophs possess genes encoding a type III SQR, comprising bacterial and archaeal SQRs of which the least is known[44].

## Discussion

In this study, we show for the first time that a microorganism can oxidize CH$_4$ and H$_2$S simultaneously, and that a methanotroph can produce biomass from CO$_2$ with H$_2$S as sole energy source. We showed that oxidation of H$_2$S is necessary because H$_2$S inhibits both respiration and CH$_4$ oxidation. Cells responded to the presence of H$_2$S by upregulating a type III sulfide:quinone oxidoreductase (SQR) and a sulfide-insensitive *ba*$_3$-type terminal oxidase (SITO). In addition, we provide evidence for an H$_2$S detoxification mechanism in methanotrophs, which, according to genomic information and the co-occurrence of methane and sulfide in a myriad of environments, seems to be widespread.

Very little is known about the effect of $H_2S$ on methanotrophy. A methanotrophic consortium sampled from a landfill showed decreased methanotrophic activity in the presence of $H_2S$[46]. In addition, $CH_4$ oxidation by *Methylocaldum* sp. SAD2, isolated from a sulfide-rich anaerobic digester, was significantly inhibited (44–60% decrease in methanol production) in the presence of 0.1% $H_2S$, but the mechanism was not explored[47,48]. '*Methylovirgula thiovorans*' strain HY1A was recently shown to be able to consume various reduced sulfur compounds together with $CH_4$, but simultaneous oxidation of $CH_4$ and $H_2S$ could not be observed[40]. In the peatland where strain HY1A was isolated from, the $H_2S$ concentration was below the detection limit, suggesting that a vigorous $H_2S$ detoxification might not be necessary. In contrast, the geothermal environments where *M. fumariolicum* SolV and other verrucomicrobial methanotrophs reside, are characterized by high concentrations of $H_2S$ (from <50 ppm to 20000 ppm)[28,35,49]. Accordingly, the demonstrated ability to fix $CO_2$ with $H_2S$ as sole energy source and efficiently oxidize $H_2S$ to $S^0$ could be highly advantageous in such harsh systems. Considering that in the natural environment multiple substrates coexist, a mixotrophic lifestyle, in which $CH_4$, $H_2$ and $H_2S$ are utilized simultaneously is expected to be more beneficial[32,50].

$H_2S$ is known to bind to metals such as copper and iron, which could lead to inhibition of the $CH_4$ oxidation capacity of the copper-dependent pMMO and terminal oxidases involved in the reduction of $O_2$[1,4–6,51,52]. Interestingly, '*Methylovirgula thiovorans*' strain HY1A only encodes an iron-dependent sMMO[40], whereas *M. fumariolicum* SolV encodes three copper-dependent pMMOs[16]. The former strain does not simultaneously oxidize $H_2S$ and $CH_4$, while the latter has a rapid $H_2S$ detoxification system to alleviate inhibition of methanotrophy. The extent to which a type of methane monooxygenase is inhibited by $H_2S$ could therefore influence the need for an $H_2S$ detoxification system. Because in *M. fumariolicum* SolV the gene encoding a type III SQR was upregulated in the presence of $H_2S$, we propose that this enzyme is responsible for the observed oxidation of $H_2S$ to elemental sulfur. Indeed, type III SQRs were shown to couple the oxidation of $H_2S$ to the reduction of quinones in several archaea and bacteria[53,54]. In verrucomicrobial methanotrophs, three different types of terminal oxidases are found: an $aa_3$-type, a $ba_3$-type, and a $cbb_3$-type[16]. Possessing multiple types of terminal oxidases allows a branched electron transport chain, which is highly advantageous in environments with fluctuating conditions and varying substrate and oxygen availability. Through respiration studies, we showed that *M. fumariolicum* SolV possesses one or more sulfide-sensitive terminal oxidases (SSTO) and at least one sulfide-insensitive terminal oxidase (SITO). Because a $ba_3$-type terminal oxidase is strongly upregulated in cells growing at high $H_2S$ loads, we propose this specific enzyme complex to be the dedicated SITO in verrucomicrobial methanotrophs. Similarly, in sulfur-grown cells of *Acidithiobacillus ferrooxidans* this $ba_3$-type oxidase was upregulated[55]. The highly upregulated heptahaem cytochrome *c* protein (MFUM_v2_1950) in *M. fumariolicum* SolV might be involved as electron carrier from SQR to the electron transport chain. This putative electron carrier could explain why $H_2S$ respiration still partially continues upon addition of methanol in the sulfide-adapted cells and not in the non-adapted cells. In the latter, the lack of this putative heptahaem electron carrier could be the limiting factor for $H_2S$ respiration, being overruled by the relatively large amounts of the electron carrier XoxGJ, mediating electron transfer from methanol to the terminal oxidase[56]. In contrast, in non-adapted cells the ratio in transcripts of the genes encoding XoxGJ and the putative heptahaem electron carrier is 27.6 compared to 1.2 in sulfide-adapted cells. Accordingly, the upregulation of the gene encoding the heptahaem electron carrier might enable sulfide respiration to occur concurrently with methanol oxidation, using the same terminal oxidase. $H_2S$ impedes both the SSTO, and the reaction catalysed by pMMO, as at 10 µM $H_2S$ the conversion of $CH_4$ was almost completely inactivated while methanol,

formate and $H_2$ conversion could still proceed. The observed decrease in $CH_4$ conversion in the chemostat at a maximum $H_2S$ load of 156 $\mu mol\ H_2S \cdot min^{-1} \cdot g\ DW^{-1}$ (liquid concentration <40 nM) was more than can be expected from our methane conversion inhibition studies and may indicate that a large portion of the respiratory chain is used for electrons generated by $H_2S$ oxidation, resulting in an overreduced Q-pool which prohibits proper functioning of alternative complex III (ACIII). Oxidation of $H_2S$ is needed to keep this molecule at low, non-inhibitory concentrations. Consequently, the electrons released from this oxidation need to be processed by the electron transport chain, leading to substrate competition during the simultaneous oxidation of multiple compounds such as $H_2S$ and $CH_4$. Similarly, it was proposed that an overactive SQR in *Rhodobacter capsulatus* could lead to an overreduction of the quinone pool[57]. The upregulated $ba_3$-type oxidase may alleviate this problem by oxidizing quinol and reducing the terminal electron acceptor $O_2$. In *Aquifex aeolicus*, a related $ba_3$-type oxidase was found in a supercomplex with SQR[58]. This terminal oxidase was shown to not only oxidize reduced cytochrome *c*, but also ubiquinol directly[59]. In strain SolV there may be an important role for the highly upregulated heptahaem protein as a dedicated electron shuttle between the quinone-accepting ACIII and the $ba_3$-type oxidase. A branched electron transport chain with different terminal oxidases enables metabolic versatility and adaptations. For example, *E. coli* uses the proton-pumping $bo_3$-type oxidase during growth but requires the sulfide-insensitive $bd$-type oxidases to keep growing in the presence of $H_2S$[60]. Interestingly, two genes are present that could encode sulfur dioxygenases (MFUM_v2_0873 and MFUM_v2_1149) to further oxidize elemental sulfur. However, the measured stoichiometry of 1 $H_2S$ to 0.48 $O_2$, the production of elemental sulfur and absence of acid production clearly show that $H_2S$ is not oxidized further to a significant extent. It remains to be investigated if methanotrophs can oxidize $H_2S$ further to sulfite and sulfate.

Cells of *M. fumariolicum* SolV were shown to rapidly oxidize $H_2S$ with a low apparent affinity constant (*i.e.*, high affinity) below 1 µM $H_2S$. The observed kinetic values are not surprising, since $H_2S$ already inhibits methanotrophy at such low concentrations. Through gas chromatography, an exact apparent affinity constant for whole cells could not be determined, as $H_2S$ consumption did not follow a typical Michaelis-Menten curve. A limitation of the respiratory capacity for $H_2S$ oxidation above about 1 µM may explain such deviation and could be resolved by purification of SQR. The observation that *M. fumariolicum* SolV reduced elemental sulfur or polysulfides to $H_2S$ in the presence of $H_2$ or methanol is intriguing. '*Methylovirgula thiovorans*' strain HY1A grown on thiosulfate increasingly produced an enzyme that resembles a protein known to possess sulfhydrogenase activity[40]. Interestingly, this enzyme clusters with the group 3b [NiFe] hydrogenase of *M. fumariolicum* SolV, thought to be involved in the production of NAD(P)H for $CO_2$ fixation[50]. Indeed, it is proposed that these hydrogenases can have sulfhydrogenase activity, which could be a mechanism to dispose of reducing equivalents[61,62]. Hence, the group 3b [NiFe] hydrogenase of *M. fumariolicum* SolV might be responsible for the conversion of $S^0$ to $H_2S$. Culturing in chemostats again turned out to be a very powerful tool to investigate the metabolism of methanotrophs[41,63]. Through adaption, *M. fumariolicum* SolV was able to respire $H_2S$ at a rate five times higher than non-adapted cells, presumably due to the upregulation of SQR and the $ba_3$-type terminal oxidase.

Verrucomicrobial methanotrophs that thrive in geothermal environments possess a clear mechanism to cope with $H_2S$. Accordingly, SQR and a sulfide-insensitive terminal oxidase could enable these methanotrophs to thrive in $H_2S$-rich environments. Indeed, pyrosequencing showed that *Methylacidimicrobium*-related 16 S rRNA gene sequences were abundantly present in the crown of concrete sewage pipes rich in $CH_4$ and $H_2S$[64]. Concerning proteobacterial methanotrophs, the effect of $H_2S$ warrants further investigation.

Because aerobic methanotrophs live in environments in which $H_2S$ is often present, we propose that the mechanism of $H_2S$ detoxification is widespread in methanotrophs in various environments.

## Methods

### Microorganism and culturing

*Methylacidiphilum fumariolicum* SolV used in this study was isolated from a mud pot of the Solfatara near Naples, Italy[28]. The genome of this strain is publicly available and accessible at Genoscope [https://mage.genoscope.cns.fr/microscope/mage/viewer.php?O_id=1176], as well as at EMBL/NCBI (BioProject PRJEA85607; accession ERS14853105). This environment is characterized by large sulfide emissions, high temperatures and extremely low pH values. The growth medium was composed of 0.2 mM $MgCl_2$, 0.2 mM $CaCl_2$, 1 mM $Na_2SO_4$, 2 mM $K_2SO_4$, 7.5 mM $(NH_4)_2SO_4$ and 1 mM $NaH_2PO_4$ and trace elements at final concentrations of 1 μM $NiCl_2$, 1 μM $CoCl_2$, 1 μM $MoO_4Na_2$, 1 μM $ZnSO_4$, 1 μM $CeCl_3$, 5 μM $MnCl_2$, 5 μM $FeSO_4$, 10 μM $CuSO_4$ and 40–50 μM nitrilotriacetic acid (NTA). Cells were grown as methane-limited continuous culture at 55 °C as described before[65], except that the pH was regulated at 2.5–3.0, that a small 400 mL chemostat was used with medium as described above and that $H_2$ was not supplemented. The oxygen concentration was regulated at 1% air saturation. In addition, a second chemostat was operated under similar conditions, but to which $H_2S$ was added through an additional gas inlet (Supplementary Fig. 1). $H_2S$ was produced by mixing 100 mM anoxic $Na_2S$ and 210 mM HCl in a 50 mL bottle with a peristaltic pump. The argon/$CO_2$ (95%/5%, v/v) gas stream to the reactor was led through this bottle. In order to determine the maximum $H_2S$ conversion rate of the chemostat, the cells were gradually exposed to higher $H_2S$ concentrations by regulating the peristaltic pump. The $H_2S$ concentrations in the gas inlet and gas outlet were determined using gas chromatography (described in the subsection: Batch incubations and gas chromatography). Because $H_2S$ was supplied through the gas inlet and therefore needs to be transferred to the liquid phase, the liquid $H_2S$ concentration will be close to or lower than its equilibrium concentration, which at 55 °C is 1.6 times the gas concentration (calculated from the Ostwald coefficient at 55 °C)[66]. In addition, to observe whether *M. fumariolicum* SolV can grow on $H_2S$ as sole energy source a fed-batch culture was operated in the same setup as the chemostat system. In this case the medium flow was stopped, and the argon/$CO_2$ gas was changed for an argon only gas stream. At the same time equal amounts of a $^{13}C$-labeled bicarbonate solution (50 mM) and HCl solution (100 mM) were additionally added to the sulfide mixing bottle, creating a $^{13}C$-$CO_2$ gas concentration of about 2%. 5 mL biomass samples from the fed-batch culture were collected by centrifugation over several days and the pellets were washed with acidified water (pH 3). Pellets were then resuspended in small amounts of acidified water and samples were subsequently pipetted into tin cups and dried overnight at 70 °C under vacuum. $^{13}CO_2$ incorporation into biomass was assessed by measuring the $^{13/12}C$ ratio using a Finnigan DeltaPlus isotope-ratio mass spectrometer (IR-MS) as described before[42].

### Membrane-inlet mass spectrometry and respiration measurements

To accurately measure dissolved gases, membrane-inlet mass spectrometry (MIMS) was performed as described previously[65], except that a 30 mL MIMS chamber was used. All rates were measured at 52 °C. The inserted probe consisted of a blunt end stainless steel tube (diameter 3 mm) that was perforated with 4–16 holes of 1 mm diameter. The holes were covered with silicon tubing (Silastic, 50VMQ Q7-4750 Dow Corning, supplied by Freudenberg Medical via VWR international, 1.96 mm outer diameter x 1.47 mm inner diameter). For easy mounting the silicon tubing was soaked briefly in hexane, which causes silicone to swell. The metal part was wetted with iso-propanol as lubricant. The probe was directly connected via a 1/8- or 1/16-inch stainless steel tube

to the MS that was operated at 40 μA emission current. Medium with a pH equal to that of the culture (pH 2.5–3.0) added to the chamber was first flushed with 3% $CO_2$ in argon gas after which the oxygen concentration was adjusted to the desired value by adding pure oxygen gas or air via the headspace. Mass 15 and 16 are both dominant masses for $CH_4$ in the mass spectrometer, but mass 15 has a much lower background signal than mass 16 and was therefore chosen to measure $CH_4$. Methane and hydrogen (mass 2) were added as a gas in the headspace or, in the case of calibration, from a saturated stock solution. These stock solutions were prepared in a closed bottle with water at room temperature and a headspace of pure gas with known pressure. For the solubility in water 1.47 mM and 0.80 mM were taken for methane and hydrogen, respectively (at 22 °C and 1 bar)[66]. When $CO_2$ production rates were to be measured, $^{13}C$-bicarbonate and equimolar amounts of sulfuric acid were added after flushing with argon. In this way the simultaneously occurring $CO_2$ fixation is mainly from $^{13}C$-labeled $CO_2$ (mass 45), leading to less interference with measurement of $CO_2$ production. At the start, unlabeled $CO_2$ (mass 44) was very low, and its increase reflected almost exclusively $CO_2$ production from unlabeled methane or methanol.

The stoichiometry of $H_2S$ oxidation was determined through pulse-wise additions of a sulfide stock solution and $O_2$ (as tiny gas bubbles with a syringe) in order to keep concentrations low at 1–20 μM $H_2S$ and 0–5 μM $O_2$. In total, 0.7–1.4 mM of $Na_2S$ was added over a period of 1.5–3 h. During this experiment, equimolar amounts of a 200 mM sulfuric acid stock solution were added simultaneously to limit the pH change within 0.2 units. The oxygen concentration was simultaneously measured in the MIMS chamber by means of a fiber-optic oxygen sensor spot (TROXSP5, PyroScience, Aachen, Germany) that was glued on the inside of the chamber. These spots could measure down to about 20 nM oxygen, which is much lower than can be measured with the mass 32 signal of MIMS.

### Batch incubations and gas chromatography

To determine kinetic parameters of $H_2S$ oxidation by sulfide-adapted cells, batch incubations were performed in 120 mL serum bottles containing 10 mL medium without any trace elements. Trace elements were omitted to minimize the effect of abiotic sulfide oxidation. The bottles were closed with butyl rubber stoppers. Incubations were performed at 55 °C and 350 rpm. $H_2S$ was prepared by mixing $Na_2S$ with HCl in a closed bottle. A volume headspace was taken and injected into 120 mL serum bottles and equilibrated for 30 min before initiating the assay by addition of cells. $H_2S$ was measured by injecting 100 μL of the headspace of the bottles with a Hamilton glass syringe into a GC (7890B GC systems Agilent technologies, Santa Clara, USA) equipped with a Carbopack BHT100 glass column (2 m, ID 2 mm) and a flame photometric detector (FPD). The areas obtained were used to calculate $H_2S$ amounts using calibration standard curves with $H_2S$. Briefly, 400 μL of a 25 mM $Na_2S$ stock (sodium sulfide nonahydrate, purity >98%, Sigma-Aldrich) was acidified with 2 mL 0.5 M HCl in a 574 mL bottle creating a headspace concentration of 17.4 nmol · $mL^{-1}$. Small volumes of the headspace were subsequently added to a 1162 mL bottle to create various $H_2S$ concentrations to be injected (0.1 mL) into the GC for calibration. The calibration curve ranged from -1 nmol · $L^{-1}$ to 1 μmol · $L^{-1}$ $H_2S$.

### RNA isolation, transcriptomics, and data analysis

For each replicate, 10 mL was sampled from the chemostat, and cells were immediately pelleted for 3 min at 15,000 × *g*, snap-frozen in liquid nitrogen and stored at −80 °C. Cells were harvested from cultures in steady state, which corresponds to constant parameters over at least 5 reactor volume changes. Total RNA was isolated using the RiboPure™ RNA Purification Kit for bacteria (Thermo Fisher Scientific, Waltham, MA, USA) according to the manufacturer's protocol. Ribosomal RNA was removed from the total RNA samples to enrich for mRNA using the

MICROBExpress™ Bacterial mRNA Enrichment Kit (Thermo Fisher Scientific) according to the manufacturer's protocol. The Qubit™ RNA HS Assay Kit (Thermo Fisher Scientific) and the Agilent RNA 6000 Nano Kit (Agilent Technologies, Waldbronn, Germany) and protocols were used for the quantitative and qualitative analysis of the extracted total RNA and enriched mRNA. The latter was used for library preparation by using the TruSeq Stranded mRNA Reference Guide (Illumina, San Diego, CA, USA) according to the manufacturer's protocol. For quantitative and qualitative assessment of the synthesized cDNA, the Qubit™ dsDNA HS Kit (Thermo Fisher Scientific) and the Agilent High Sensitivity DNA kit (Agilent Technologies) and protocols were used. Transcriptome reads were checked for quality using FastQC[67] and subsequently trimmed 10 base pairs at the 5' end and 5 base pairs at the 3' end of each read. Reads were mapped against the *M. fumariolicum* SolV complete genome (accession number LM997411)[68] using Bowtie2[69]. The remainder of the analysis and the production of images was performed in version 4.0.2 of the R environment[70]. The mapped read counts per gene were determined using Rsubread[71] and fold change and dispersion were estimated using DEseq2[72]. Before doing any statistics, principal component analysis on the top 1000 genes by variance of each sample was performed to check whether samples within the same condition were both similar to each sample part of the same condition, and dissimilar to any other sample. For differential expression, a Wald test was employed by DEseq2 to calculate adjusted *p*-values. Differences in counts were considered to be significant if the basemean was >4, the $\log_2$-fold change was higher than [0.58] and the adjusted *p*-value was ≤0.05. For easy comparisons between samples, TPM (Transcripts Per Kilobase Million) values were calculated.

### TOC measurements

The total organic carbon (TOC) concentrations of the cultures were determined using a TOC-L CPH/CPN analyzer (Shimadzu, Duisburg, Germany). Samples were diluted three times in Milli-Q water before measurements and subsequently sparged for 20 min with ozone while stirring to remove $CO_2$ from the liquid. Acidification of the solutions was not needed due to the low pH of the samples. An optical density of 1 measured at 600 nm is equivalent to ~450 mg dry weight (DW) per litre.

### Phylogenetic analysis

All available genome sequences of known methylotrophs from the orders Methylococcales (Gammaproteobacteria) and Methylacidiphilales (Verrucomicrobia), the families *Methylocystaceae* and *Beijerinckia* (Alphaproteobacteria), and the genus *Methylomirabilis* were retrieved from the NCBI database. Genomes of methanotrophs were selected by blasting amino acid sequences of PmoA from *Methylococcus capsulatus* (SwissProt accession Q607G3) and sMMO from *Methylosinus acidophilus* (NCBI accession AAY83388.1) with an e-value threshold of $10^{-3}$ and a %-id threshold of >30%. Genomes containing a methane monooxygenase sequence were subsequently mined for putative SQR sequences by blasting a representative sequence of each of the SQR subtypes as defined by previous research[44]: type I, WP_010961392.1; type II, WP_011001489.1; type III, WP_009059890.1; type IV, WP_011372252.1; type V, WP_012502121.1; type VI, WP_011439951.1. Putative SQR sequences were aligned with those in the phylogenetic tree of[44] using Muscle 3.8.1551[73] with default settings. A maximum-likelihood phylogenetic tree with 500 bootstrap replicates was constructed using RAxML 8.2.10[74] using the rapid bootstrapping option and the LG amino acid substitution model[75]. The final tree was visualized using MEGA7 and the clade of flavocytochrome *c* sulfide dehydrogenase (FCSD) sequences was used as outgroup.

### Reporting summary

Further information on research design is available in the Nature Portfolio Reporting Summary linked to this article.

## Data availability

The RNA sequencing data in this study have been deposited in the NCBI database under accession number PRJNA766544. The genome of *Methylacidiphilum fumariolicum* SolV has been deposited in the NCBI database under accession number ERS14853105. Supplementary Data 1 and the Source Data file are also available on figshare (https://doi.org/10.6084/m9.figshare.22779005). Source data are provided with this paper.

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

## Acknowledgements

R.A.S., S.S.M., T.B., and H.J.M.O.d.C. were supported by the European Research Council (ERC Advanced Grant project VOLCANO 669371), M.S.M.J. was supported by the European Research Council (ERC Advanced Grant project EcoMoM 339880), W.V. was supported by the Netherlands Organisation for Scientific Research (NWO) grant VI.Vidi.192.001 and S.H.P. was supported by the Netherlands Organisation for Scientific Research (NWO) grant ALWOP.308. We want to thank Dr. Marianne Guiral and Dr. Frauke Baymann (CNRS, Aix-Marseille University, Marseille, France) for fruitful discussions. The LABGeM (CEA/Genoscope & CNRS UMR8030), the France Génomique and French Bioinformatics Institute national infrastructures (funded as part of Investissement d'Avenir program managed by Agence Nationale pour la Recherche, contracts ANR-10-INBS-09 and ANR-11-INBS-0013) are acknowledged for support within the MicroScope annotation platform.

## Author contributions

R.A.S., S.S.M., A.P. and H.J.M.O.d.C. designed the project and experiments. R.A.S., S.S.M., T.v.E., C.A.I., T.v.A., W.V. and A.P. conducted the experiments. T.B. conducted phylogenetic analyses. R.A.S., S.H.P., S.S.M., T.v.E., C.A.I. and A.P. performed data analyses. R.A.S., M.S.M.J., H.J.M.O.d.C. and A.P. wrote the manuscript. R.A.S., M.S.M.J., H.J.M.O.d.C. and A.P. supervised the research.

## Competing interests

The authors declare no competing interests.
