## [Peer Review File · Nature Communications]

Simultaneous sulfide and methane oxidation by an extremophileReviewer #1 (Remarks to the Author):

The manuscript reports on the interesting finding that *Methylacidiphilum fumarolicum* SolV oxidizes methane and sulfide at the same time. This is shown mainly by experiments in continuous culture. The finding that sulfur compounds and methane are simultaneously oxidized is not new. Here, the new finding is that it is sulfide itself that is oxidized although this compound inhibits methane oxidation and oxygen respiration. The authors furthermore show that for the methanotroph oxidation of hydrogen sulfide allows production of biomass from carbon dioxide in the absence of methane. Sulfide oxidation in the presence of methane appears to be a pure detoxification process and leads to accumulation of elemental sulfur as the product.

As a whole the idea and conclusions of the project are interesting but I see some problems with the writing style and also with experimental design.

First, very many detailed numbers are listed in the main text regarding substrate conversion rates or respiration rates. Many of these numbers can better be provided in a Table.

The major problem for me with the manuscript is the part on gene regulation. First, the locus tags given in the text and also in table 2 and in supplementary file 1 are not the same locus tags as used in the publicly available complete genome for the organism (acc no LM997411 as listed by the authors). It is thus absolutely impossible for a reviewer (and also for a later reader), to verify the authors' conclusions. This needs to be urgently changed and the locus tags available at NCBI need to be integrated into the table. In addition, I cannot confirm that the authors mentioned all genes relevant to sulfide oxidation. According to my own analysis there is a second sulfide:quinone oxidoreductase, SqrV (MFUM_RS00250) and two sulfur dioxygenases (SDOIII) (MFUM_RS02615 and MFUM_RS07615). In principle, SQR and SDO together can produce sulfite. As I cannot link the NCBI locus tags to the authors' tags, it is impossible to see whether transcripts are also more abundant for the genes not mentioned. Which other genes are more abundantly transcribed? A less than twofold higher abundance of transcripts as observed for *sqr* (mfumv2_0219) does not appear very convincing to me.

What was the selection criterion for the genes presented in Table 2? Which other genes were very strongly affected?

It is my strong view that this kind of information should be made available before a final assessment can be made as to whether the manuscript is suitable for publication.

Reviewer #2 (Remarks to the Author):

This is an excellent manuscript that describes the simultaneous use of hydrogen sulfide and methane by the thermophilic and acidophilic methanotroph *Methylacidiphilum fumarolicum* SolV. Methanotrophs are often present in environments which are rich in both methane and inorganic sulfur compounds, and it has been known for many years that these sulfur compounds and in particular hydrogen sulfide can be toxic to and inhibit growth of aerobic methanotrophs. The sorts of environments where *M. fumarolicum* thrive, geothermal ecosystems such as fumaroles, therefore, seem like good places to investigate the underlying mechanisms by which these methanotrophs can overcome hydrogen sulfide toxicity. *M. fumarolicum* has already been shown to be quite a metabolically versatile methanotroph, growing well in harsh environments and is therefore a good model organism for this study. Recently *Methylovirgula thiovorans* has been demonstrated to grow on a range of different carbon compounds and a number of sulfur sources, including thiosulfate, tetrathionate and elemental sulfur but not with the more toxic hydrogen sulfide.

This manuscript therefore represents a significant advance in the field and demonstrates the important and exciting links between carbon and sulfur cycling in the environment. The authors use a series of elegant microbial physiology methods including chemostat culture. This study illustrates how effective continuous culture methods (sadly not used as much as they should be in microbiology these days) can be in rigorously defining and monitoring growth conditions. Their work clearly shows that *M. fumarolicum* can oxidize methane and hydrogen sulfide simultaneously and that it can make biomass from carbon dioxide using hydrogen sulfide as the only energy source. They also provide, using some careful transcriptomics experiments, gene regulation evidence to support a clear mechanism by which *M. fumarolicum* responds to the presence of hydrogen sulfide by increasing expression a sulfide:quinone oxidoreductase (SQR) together with a

sulfide-insensitive terminal oxidase (ba3 type). Thus providing a very plausible way in which hydrogen sulfide could be detoxified by methanotrophs. The distribution of SQRs in methanotrophs is also demonstrated by surveying the genomes of extant methanotrophs. Overall a very impressive body of work. The experiments have been well-thought out and rigorously executed. The manuscript is well-written and the figures, tables and supplementary information is all laid out in a very clear fashion. I enjoyed reading this work. It is likely to be highly cited in the future and opens up new avenues of investigation into methanotroph biology and their roles in carbon and sulfur cycling in the environment.

I have no major issues with the manuscript. A few minor points for consideration:

Line 258. I assume these cultures were in steady-state at the time of harvesting? How was steady-state defined (constant parameters over how many volume changes?)?

Table 2: was there significant up-regulation or down-regulation of methanol oxidation genes ?

Lines 289-294. In which genera of methanotrophs were SQRs notably absent?

Figure 3: It would be nice to include *Methylovirgula thiovorans* strain HY1 in this phylogenetic tree.

Line 351: see earlier comment also. Relatively large amounts of XoxJG- do these levels respond to the presence of hydrogen sulfide?

Reviewer #3 (Remarks to the Author):

Simultaneous sulfide and methane oxidation by an extremophile

Rob A. Schmitz et al.

Methanotrophs play an important role in mitigating methane emissions to the atmosphere, and since hydrogen sulfide is often present together with methane, particularly in extreme environments, the effect on methanotrophs is both interesting and environmentally relevant. This manuscript continues this group's exploration of the diverse metabolism of the extremophile verrucomicrobial methanotroph, *Methylacidiphilum fumarolicum* SolV. The manuscript reflects a considerable amount of experimental work, is no doubt based on the solid and innovative science for which this lab is renowned, and is mostly extremely well written and concise. However, the manuscript is complicated and not too easy to follow and I have some comments on the presentation of the data which I hope can be relatively easily addressed.

L148 There are a lot of data presented, and it would help if the authors indicated where the figure of 0.5 μM comes from. Similarly, L 149 and Figure S4, I'm not sure how the figures of % inhibition were derived. It's a shame we can't see the rates of CH_4 uptake at the same concentrations with variable $[\text{H}_2\text{S}]$. Looking at the figure, when methane was re-added (red arrow) the rate seems higher than 80% of the T0 figure – is this due to depletion of H_2S , or how is this explained? When H_2S was spiked (blue arrow) the $[\text{CH}_4]$ was quite low – can we be sure that this is saturating? Please clarify or elaborate.

L174 How was the 1:0.48 ratio quantified?

L175 – 189 This section does not seem to be described in methods, and perhaps a figure in Supplementary would help.

L193. "Increased up to 50%" – do the authors mean "increased by 50%", otherwise please indicate 50% of what.

L 192 – 203 and 217 - 224. These sections are very confusing. Please indicate where these data are presented. I could not find any figure or table showing the cessation or reduction of H_2S consumption together with increase of respiration when methanol or hydrogen or formate was added. 20 – 40 μM sulfide does not seem to match with the concentrations given in Table 1 (10 – 20 and 40 – 80 μM). L201 "as opposed to methane" – where is this shown? My understanding is that Table 1 refers to cells removed from the two growth conditions and tested with the listed substrates in the MIMS chamber, whereas these text sections seem to refer to more complicated experiments in which more than one substrate was added sequentially. Unless I am misunderstanding completely, these sections refer to experiments which are not described or presented. Please clarify. I suggest a figure showing these changes in consumption, and referring to it in the text.

Table 1. Caption, "Respiration refers to O_2 consumption of the listed substrates" – should this be "in response to the listed substrates"?

Table 1 What is meant by "CH4 respiration" under maximum rates? This figure seems to be arrived at by subtracting "CH4 conversion" from "O2 for CH4", and therefore, if I'm understanding correctly, is the expected O2 consumption for the rest of the CH4 consumption pathway, following the initial oxidation (?) in which case the term CH4 respiration is confusing.

L212 Consider prefacing "Due to" with "Likely"

L222. Mostly the H2S concentrations are given as a range, should this be 20 – 40 μ M?

L226 – 228. I did not understand this sentence and I'm not sure how a terminal oxidase can have a preference for a substrate as such. Please consider if this sentence is appropriate and/or clarify.

L237 – 238 If the authors are claiming that the difference here is due to high [O2] and low [H2S] please give the O2 concentration and identify the rate in the MIMS chamber. From Table 1, unless I'm mistaken, this seems to be 100 +/- 11, so the rate is more than slightly higher. I don't fully understand the phrase "partially alleviating inhibition" (L241) – inhibition of what and by what, and how does [O2] play a role? Please rephrase and/or clarify.

L247 Should this be sequential addition of O2?

Fig 2a Please mention in the legend that the reactions were initiated by addition of cells and at which timepoints.

L264 Since several mentions of SQR types are made throughout, it would be helpful to include, either in the intro or in the section commencing L285, a sentence or two describing the basis of SQR type classification and what is known about the properties of the different groups. If space is limited, L42 – 45 could be omitted, since this passage about the harmful effects of H2S is largely irrelevant in the context of this study.

L274 H2S or sulfide? Please give a few examples of the metabolites.

L277 Delete "their"

L278 conversation >> conversion

Table 2 Why not use the same colors as Fig S7?

L289 "different types" unnecessarily vague and emphasises the value of definition of SQR types

Fig 3 This tree is not very informative as is. I suggest identifying the characterised S-oxidizers.

Define FCSD which has not yet been done. Specify amino acid or nucleotide and define scale bar.

L311 delete "much more" or define more than what

L346 Please identify the highly upregulated gene (mfumv2_1950?)

L351 – 355 Please supply a reference that these two polypeptides (XoxJG) function together as an electron carrier. Please indicate the extent of the change in ratio and clarify that this is gene transcripts, not protein (as suggested in the text).

L360 – 362 please rephrase in more accurate and scientific language

L376 High affinity constant = low affinity

L421 – 2 and L482 If molarity is given, we don't need the water of hydration

L446 Please indicate the amount of cells in the chamber

L450 shortly >> briefly

L462 insert "measurement of" after "interference with" if that is what is meant

L482 Define the calibration curve range. HCl added?

In many places the tense is confusing. To report experimental results the past tense should be used. For example, L215, as was the case for CH4... L307, cells responded to the presence...

REBUTTAL TO THE REVIEWER COMMENTS

Reviewer #1 (Remarks to the Author):

The manuscript reports on the interesting finding that *Methylacidiphilum fumarolicum* SolV oxidizes methane and sulfide at the same time. This is shown mainly by experiments in continuous culture. The finding that sulfur compounds and methane are simultaneously oxidized is not new. Here, the new finding is that it is sulfide itself that is oxidized although this compound inhibits methane oxidation and oxygen respiration. The authors furthermore show that for the methanotroph oxidation of hydrogen sulfide allows production of biomass from carbon dioxide in the absence of methane. Sulfide oxidation in the presence of methane appears to be a pure detoxification process and leads to accumulation of elemental sulfur as the product. As a whole the idea and conclusions of the project are interesting but I see some problems with the writing style and also with experimental design.

First, very many detailed numbers are listed in the main text regarding substrate conversion rates or respiration rates. Many of these numbers can better be provided in a Table.

We want to thank the reviewer for the suggestion and agree that some rates are better listed in a Table than in text. Most conversion and respiration rates are moved from the main text to the table and figures. In addition, we have created additional figures (Figures 2, 3 and 4, and Supplementary Figures 4, 5, 6 and 7) to visualize the rates, so that the reader can see how they were derived. Thank you for your suggestion, we hope the manuscript is easier to read now.

The major problem for me with the manuscript is the part on gene regulation. First, the locus tags given in the text and also in table 2 and in supplementary file 1 are not the same locus tags as used in the publicly available complete genome for the organism (acc no LM997411 as listed by the authors). It is thus absolutely impossible for a reviewer (and also for a later reader), to verify the authors' conclusions. This needs to be urgently changed and the locus tags available at NCBI need to be integrated into the table.

We apologize for not clearly stating where the genome we used for transcriptomic analyses is publicly available. The genome present in Genbank (LM997411) was published and submitted by us in 2014 (Anvar et al.; doi: 10.1186/1471-2164-15-914). We re-analysed, updated and re-annotated this genome in detail using the Microscope platform at Genoscope (<https://mage.genoscope.cns.fr/microscope>). The genome is publicly available and accessible at Genoscope under the name *Methylacidiphilum fumarolicum* SolV (chromosome MFUM.2; ID = 1176). We created a hyperlink leading to this genome in the method section (Line 449). This re-annotated genome is the genome version with locus tags used in this study. We have now also uploaded this updated genome to EMBL/NCBI (BioProject PRJEA85607; accession ERS14853105).

In addition, I cannot confirm that the authors mentioned all genes relevant to sulfide oxidation. According to my own analysis there is a second sulfide:quinone oxidoreductase, SqrV (MFUM_RS00250) and two sulfur dioxygenases (SDOIII) (MFUM_RS02615 and MFUM_RS07615). In principle, SQR and SDO together can produce sulfite. As I cannot link the NCBI locus tags to the authors' tags, it is impossible to see whether transcripts are also more abundant for the genes not mentioned.

We thank the reviewer for the genome analysis. To compare the abundances of gene transcripts, all raw transcriptomics data (including counts and TPM values) can be viewed the Excel File "Supplementary File 1". In addition, we have listed the upregulated and downregulated genes of sulfide-adapted versus non-adapted cells. All amino acid sequences and their locus tags are also stated in this file.

The gene MFUM_RS00250 (MFUM_v2_0138) mentioned by the reviewer encodes for the following amino acid sequence:

```
>WP_009058074.1 NAD(P)/FAD-dependent oxidoreductase [Methylacidiphilum
fumariolicum]
MRPFHVIGGGGFGGLTAAKTLGKEIQKRKLPCKLTLIDKENHHLFQPLLYQVATAGLAATDIAVPIRSI
LSRIQEVEVRMETIERIDLEKKMLFCSKGVLSYDFLILSLGMRVNYFGHEEWSSFCLGLKTLGDGRSIRN
VILNAFEKAEIETNPKEREKYMTFVIVGGGPTGVELAGALAELESKKALKKDFRNIDPANTRII LLEAAPR
ILLSYNERLSALARSRLKMGVEIMVSKPVEKIEKGKIFYKGGMIEASTILWAAGVCAMDLPGLDVPKVK
DGRIVLEDLTVPGHPEIFVIGDMAMVPGVPAVAPAAIQMGKYAAAYEISRRVACQVGRRTGRDFLKPRAF
HYFDKGMMTTLGRGKAI VQFHNFGFNGYLAWI VLLVHIITLISFRNRLTVLIQWAWAYFRFKPGARLLS
K
```

This protein is annotated as NAD(P)/FAD-dependent oxidoreductase and classified into the protein family “Alternative NADH dehydrogenase” (IPR045024) by InterPro. Blasting MFUM_RS00250 (MFUM_v2_0138) against the NCBI database reveals almost exclusively high-confidence hits against other NAD(P)/FAD-dependent oxidoreductases. However, we are aware that the same is true for SQR (MFUM_v2_0219) mentioned in our study.

We dove a bit deeper into the classification of SQRs. Marci et al. (2009; 10.1002/prot.22665) provide a classification of SQRs into six groups, with “fingerprint regions” based on the well-studied SQR (<https://www.uniprot.org/uniprotkb/Q7ZAG8/>) of *Acidianus ambivalens*. This SQR of *A. ambivalens* aligns well with SQR (MFUM_v2_0219) found in *M. fumariolicum* SolV with 24.38% identity with 95% query cover (E value: 5e-17). Based on Marci et al. (2009), MFUM_v2_0219 can be classified as Type III SQR.

Aligning MFUM_RS00250 (MFUM_v2_0138) mentioned by the reviewer with SQR of *A. ambivalens* shows 25.00% identity, but only over a small part of the sequence (24% query cover, E value: 3e-06). Searching for “fingerprint regions” is complicated due to the low query cover and a manual search suggests it might be part of the group VI SQRs. We do agree with the reviewer that it could be an SQR but is it difficult to conclude based on gene comparisons, especially because SQRs are notoriously difficult to compare due their low sequence identities.

We compared the TPM values of the gene encoding the SQR mentioned in our study (MFUM_v2_0219) and the gene encoding the SQR mentioned by the reviewer (MFUM_v2_0138) in cells grown in the combined methane-sulfide reactor (sulfide-adapted cells) versus cells grown in the methane reactor (non-adapted cells):

MFUM_v2_0138

TPM: 30 ± 3 (methane reactor)

TPM: 35 ± 5 (methane-sulfide reactor)

MFUM_v2_0219

TPM: 210 ± 7 (methane reactor)

TPM: 479 ± 79 (methane-sulfide reactor)

Based on transcriptomic analyses, we determined that the gene MFUM_v2_0219 is transcribed to a greater extent than MFUM_v2_0138 and that MFUM_v2_0219 is significantly upregulated when grown under both methane and sulfide, whereas MFUM_v2_0318 is not. We again want to thank the reviewer for the genetic analyses, and we decided to add the following information to the manuscript: “Based on gene comparisons, a second gene (Mfum_v2_0138) might encode an SQR, although this gene is not significantly upregulated in the presence of H₂S and transcribed to a much smaller degree than Mfum_v2_0219 in sulfide-adapted cells (**Supplementary File 1**).”

We would like to thank the author for scanning the genome for putative SDO genes. MFUM_RS02615

(MFumv2_1149) and MFUM_RS07615 (MfumV2_0873) are both annotated as Zn-dependent hydrolase, glyoxylase family. Recently, a novel sulfur dioxygenase in *Acidithiobacillus caldus* (gene A5904_0421; NCBI GenBank accession number OAN04050.1) was shown to possess SDO activity. Blasting this protein against the genome of strain SolV indeed gives two hits:

MFUM_v2_0873 [Methylacidiphilum fumariolicum SolV chromosome MFUM] gloB | Zn-dependent hydrolase, glyoxylase family
Length=271 Score = 60.5 bits (145), Expect = 5e-12 Identities = 47/173 (27%), Positives = 79/173 (46%), Gaps = 27/173 (16%)

MFUM_v2_1149 [Methylacidiphilum fumariolicum SolV chromosome MFUM] gloB | Zn-dependent hydrolase, glyoxylase family | automatic/finished Length=211 Score = 51.2 bits (121), Expect = 5e-09 Identities = 53/215 (25%), Positives = 83/215 (39%), Gaps = 46/215 (21%).

We aligned SDO of *A. caldus* with MFUM_v2_0873, which resulted in a 71% query coverage and 26.86% identity (E value 5e-15). Aligning SDO of *A. caldus* with MFUM_v2_1149 resulted in a 79% query coverage and 25.00% identity (E value 4e-13).

We compared the TPM values of two genes mentioned by the reviewer (MFUM_v2_0873 and MFUM_v2_1149) in cells grown in the combined methane-sulfide reactor (sulfide-adapted cells) versus cells grown in the methane reactor (non-adapted cells):

MFUM_v2_0873
TPM: 457 ± 64 (methane reactor)
TPM: 554 ± 54 (methane-sulfide reactor)

MFUM_v2_1149
TPM: 56 ± 16 (methane reactor)
TPM: 65 ± 12 (methane-sulfide reactor)

Clearly, MFUM_v2_0873 is transcribed to a much larger extent than MFUM_v2_1149. Based on gene comparisons, the genes mentioned by the reviewer could indeed be involved in sulfur oxidation to sulfite.

We deleted the sentence “An enzymatic pathway from elemental sulfur to sulfite (SO₃²⁻) could not be resolved, but sulfite might also be produced chemically.” (Line 259-261). We also deleted the sentence in the discussion stating “downstream genes involved in the oxidation of elemental sulfur to sulfite are seemingly absent in strain SolV”.

We have added the text: “Two genes (MFUM_v2_0873 and MFUM_v2_1149) are transcribed that might encode sulfur dioxygenases, which could putatively oxidize elemental sulfur to sulfite (SO₃²⁻) (**Supplementary File 1**).” ... “In sulfide-adapted cells, the putative sulfur dioxygenase (MFUM_v2_0873) is transcribed to a similar degree as SQR (MFUM_v2_0219). However, based on the stoichiometry of 1 H₂S : 0.48 O₂ (± 0.005; *n* = 3) quantified in the MIMS chamber, the conversion of elemental sulfur and polysulfides via sulfite to sulfate is thought to have a minor role under the tested conditions. In addition, the oxidation of H₂S was never accompanied by a decrease in pH, which would have been the case if elemental sulfur had been oxidized further to thiosulfate, sulfite or sulfate.”

In the discussion section, we have deleted the sentence “In line with this, downstream genes involved in the oxidation of elemental sulfur to sulfite are seemingly absent in strain SolV. Whether genes involved in this pathway were lost during evolution or never acquired by verrucomicrobial methanotrophs remains to be investigated.” Instead, we now

state: “Interestingly, two genes are present that could incode sulfur dioxygenases (MFUM_V2_0873 and MFUM_v2_1149) to further oxidize elemental sulfur. However, the measured stoichiometry of 1 H₂S to 0.48 O₂, the production of elemental sulfur and absence of acid production clearly show that H₂S is not oxidized further to a significant extent. It remains to be investigated if methanotrophs can oxidize H₂S further to sulfite and sulfate.”

Which other genes are more abundantly transcribed?

How abundantly genes are transcribed is shown in Supplementary File 1. The higher the basemean value (the mean of the nomalized counts of all samples), the more the gene is transcribed. For convenience, we have added an extra sheet called “Amino acid sequences” that gives the amino acid sequences encoded by all the list genes, together with the (putative) gene products, as given through a combination of automated and manual annotation.

A less than twofold higher abundance of transcripts as observed for *sqr* (mfumv2_0219) does not appear very convincing to me. What was the selection criterion for the genes presented in Table 2?

In Table 2, upregulated and downregulated genes that are relevant to the oxidation of sulfur compounds, the oxidation of methane, the respiratory chain and assimilatory sulfide production are listed. The selection criteria for genes presented in Table 2 are a basemean higher than 4, an upregulation or downregulation factor higher than 1.5 and an adjusted p-value ≤ 0.05 (all averages of triplicates). The basemean is the mean of the normalized counts of all samples (this calculation normalizes for sequencing depth).

We agree that a twofold higher abundance of transcripts is not a large difference. In fact, we expected a more or less constitutive expression of SQR because we showed that sulfide already inhibits methane oxidation at low concentration. Upregulation is limited because the non-adapted culture already has a rather high sulfide-oxidizing capacity. The fact that SQR is not very highly upregulated may point towards SQR not being the limiting factor. It might be the highly upregulated gene encoding the heptahaem cytochrome *c* protein that is the principal limitation, but more research is needed to investigate the role of this protein. Altogether, in an acidic environment in which H₂S is typically present (mimicked in chemostats in our study), we expected the gene encoding SQR to always be “switched on”, to rapidly reactor to the sudden presence of H₂S.

Cut-off values of upregulation and downregulation factors used in transcriptomics are to a certain extent arbitrarily chosen. We chose to use the statistically robust method *DESeq2* for differential analyses of count data. What makes this result significant is not the absolute difference in expression, but the variance in expression relative to other genes expressed at a similar level. This method for estimating variance-mean dependence, as employed by us through *DESeq2*, is an excellent technique to detect differentially expressed genes by sidestepping and ameliorating the common problem that genes with low or high expression levels are sensitive to Type I and II errors, respectively.

Which other genes were very strongly affected? It is my strong view that this kind of information should be made available before a final assessment can be made as to whether the manuscript is suitable for publication.

Other genes that were affected by the cultivation in the presence of sulfide, are listed in Supplementary File 1. In these lists, the annotation is given to inform the reader about the (putative) function of the gene. The selection criteria for genes listed as “upregulated” or “downregulated” are a basemean higher than 4, an upregulation or downregulation factor higher than 1.5 and an adjusted p-value ≤ 0.05 (all averages of triplicates).

Reviewer #2 (Remarks to the Author):

This is an excellent manuscript that describes the simultaneous use of hydrogen sulfide and methane by the thermophilic and acidophilic methanotroph *Methylacidiphilum fumarolicum* SoIV. Methanotrophs are often present in environments which are rich in both methane and inorganic sulfur compounds, and it has been known for many years that these sulfur compounds and in particular hydrogen sulfide can be toxic to and inhibit growth of aerobic methanotrophs. The sorts of environments where *M. fumarolicum* thrive, geothermal ecosystems such as fumaroles, therefore, seem like good places to investigate the underlying mechanisms by which these methanotrophs can overcome hydrogen sulfide toxicity. *M. fumarolicum* has already been shown to be quite a metabolically versatile methanotroph, growing well in harsh environments and is therefore a good model organism for this study. Recently *Methylovirgula thiovorans* has been demonstrated to grow on a range of different carbon compounds and a number of sulfur sources, including thiosulfate, tetrathionate and elemental sulfur but not with the more toxic hydrogen sulfide. This manuscript therefore represents a significant advance in the field and demonstrates the important and exciting links between carbon and sulfur cycling in the environment. The authors use a series of elegant microbial physiology methods including chemostat culture. This study illustrates how effective continuous culture methods (sadly not used as much as they should be in microbiology these days) can be in rigorously defining and monitoring growth conditions. Their work clearly shows that *M. fumarolicum* can oxidize methane and hydrogen sulfide simultaneously and that it can make biomass from carbon dioxide using hydrogen sulfide as the only energy source. They also provide, using some careful transcriptomics experiments, gene regulation evidence to support a clear mechanism by which *M. fumarolicum* responds to the presence of hydrogen sulfide by increasing expression a sulfide:quinone oxidoreductase (SQR) together with a sulfide-insensitive terminal oxidase (ba3 type). Thus providing a very plausible way in which hydrogen sulfide could be detoxified by methanotrophs. The distribution of SQRs in methanotrophs is also demonstrated by surveying the genomes of extant methanotrophs. Overall a very impressive body of work. The experiments have been well-thought out and rigorously executed. The manuscript is well-written and the figures, tables and supplementary information is all laid out in a very clear fashion. I enjoyed reading this work. It is likely to be highly cited in the future and opens up new avenues of investigation into methanotroph biology and their roles in carbon and sulfur cycling in the environment.

We would like to thank the reviewer for the very kind words regarding our study and the appreciation for the methods that we have developed and used in this study.

I have no major issues with the manuscript. A few minor points for consideration:

Line 258. I assume these cultures were in steady-state at the time of harvesting? How was steady-state defined (constant parameters over how many volume changes?)?

These cultures were indeed in steady-state at the time of harvesting. The steady-state was defined by constant parameters over at least 5 reactor volume changes. We have stated in line 271 that the cultures were in steady state, and added to the method section (Line 522) that this corresponds to constant parameters over at least 5 reactor volume changes.

Table 2: was there significant up-regulation or down-regulation of methanol oxidation genes?

There was no significant upregulation or downregulation of the XoxF-type methanol dehydrogenase (Mfum_v2_1183; Supplementary File 1). We compared the TPM values of the gene encoding XoxF (Mfum_v2_1183) in cells grown in the combined methane-sulfide reactor (sulfide-adapted cells) versus cells grown in the methane reactor (non-adapted cells):

Mfum_v2_1183

TPM: 6225 ± 464 (methane reactor)

TPM: 5533 ± 437 (methane-sulfide reactor)

Indeed, there seems to be a small regulation, but not significant under the selection criteria (a basemean higher than 4, an upregulation or downregulation factor higher than 1.5 and an adjusted p-value ≤ 0.05 (all averages of triplicates). Apparently, exposure of cells to sulfide under the conditions used in this study leads to a significant downregulation of the methane monooxygenase, not the methanol dehydrogenase.

Lines 289-294. In which genera of methanotrophs were SQRs notably absent?

SQRs seems to be widespread in microbes that possess a gene encoding pMMO or sMMO (or both). SQR is found in alphaproteobacterial and gammaproteobacterial methanotrophs, verrucomicrobial methanotrophs and *Methylomirabilis* methanotrophs. As exception: blast searching against the genomes of the genera *Methylosoma* and *Methyloparacoccus* did not lead to significant hits. However, it could be that SQRs are present, but that they are not identical enough and hence do not lead to a significant blast hit (SQRs are notoriously difficult to compare due to their low identity). In addition, the vast majority of SQRs in methanotrophs are (imperfectly) annotated as FAD-dependent oxidoreductases. Altogether, the conclusion we make in our study is that SQR is widespread among aerobic methanotrophs, and we sincerely hope our findings will advance the field to study sulfide oxidation in methanotrophs.

Figure 3: It would be nice to include *Methylovirgula thiovorans* strain HY1 in this phylogenetic tree.

Thank you for your comment. We have included this strain in the phylogenetic tree. Due to limited space, we have decided to place a large phylogenetic tree in the Supplementary Information instead, in which *Methylovirgula thiovorans* strain HY1 is included.

Line 351: see earlier comment also. Relatively large amounts of XoxJG-do these levels respond to the presence of hydrogen sulfide?

The gene (Mfum_v2_1185) encoding the fusion protein XoxGJ is not significantly upregulated or downregulated in the presence of sulfide (Supplementary File 1). We compared the TPM values of the gene encoding XoxGJ (Mfum_v2_1185) in cells grown in the combined methane-sulfide reactor (sulfide-adapted cells) versus cells grown in the methane reactor (non-adapted cells):

Mfum_v2_1185

TPM: 534 ± 91 (methane reactor)

TPM: 448 ± 74 (methane-sulfide reactor)

In addition, we know that the protein cytochrome *c* protein XoxGJ is present at high concentrations in the cells, because its purification could actually be tracked by looking at the yellow colour (typical for this protein) eluting from the ion-exchange columns (Versantvoort et al., 2019; <https://doi.org/10.1016/j.bbapap.2019.04.001>).

Reviewer #3 (Remarks to the Author):

Simultaneous sulfide and methane oxidation by an extremophile Rob A. Schmitz et al.

Methanotrophs play an important role in mitigating methane emissions to the atmosphere, and since hydrogen sulfide is often present together with methane, particularly in extreme environments, the effect on methanotrophs is both interesting and environmentally relevant. This manuscript continues this group's exploration of the diverse metabolism of the extremophile verrucomicrobial methanotroph, *Methylacidiphilum fumarolicum* SolV. The manuscript reflects a considerable amount of experimental work, is no doubt based on the solid and innovative science for which this lab is renowned, and is mostly extremely well written and concise. However, the manuscript is complicated and not too easy to follow and I have some comments on the presentation of the data which I hope can be relatively easily addressed.

We would like to thank the reviewer for the compliments and the excellent suggestions to make the manuscript easier to follow. Please see below how we have addressed your comments and suggestions.

L148 There are a lot of data presented, and it would help if the authors indicated where the figure of 0.5 uM comes from. Similarly, L 149 and Figure S4, I'm not sure how the figures of % inhibition were derived. It's a shame we can't see the rates of CH₄ uptake at the same concentrations with variable [H₂S]. Looking at the figure, when methane was re-added (red arrow) the rate seems higher than 80% of the T0 figure – is this due to depletion of H₂S, or how is this explained? When H₂S was spiked (blue arrow) the [CH₄] was quite low – can we be sure that this is saturating? Please clarify or elaborate.

Thank you for your good suggestion. Because the detection limit of the MIMS for H₂S is around 1 μM, we have decided to state that the cells are affected by a concentration as low as 1 μM, instead of 0.5-1 μM.

We have replaced the previous figure with a figure showing an additional experiment, in which we show inhibition of methane consumption at variable sulfide concentrations (Figure 2). To make the text better readable, we have now visualized the data and rates in the figure. Accordingly, the reader can now see how the inhibitions percentages were derived. The experiment was performed at methane concentrations of 30 μM or higher, to ensure maximum methane consumption velocities (*i.e.*, the methane consumption rates are not affected by the methane concentration, as the apparent affinity constant for methane is 6 μM, which we showed in 2007: <https://doi.org/10.1038/nature06222>). The non-adapted cells (as well as sulfide-adapted cells) were affected by an H₂S concentration as low as 1 μM. CH₄ oxidation was inhibited by about 25%, 75-85% and 95% in the presence of 2 μM, 4-5 μM and 10 μM H₂S, respectively.

L174 How was the 1:0.48 ratio quantified?

In the MIMS chamber, cells of *M. fumarolicum* SolV were exposed to pulse-wise additions of sulfide and oxygen (described in the materials and methods section, under the subheading “Membrane-inlet mass spectrometry and respiration measurements”). In brief, consumption of H₂S was measured using MIMS while simultaneously O₂ consumption was measured using a fiber-optic sensor spot. In the results section, we have now stated “a H₂S:O₂ stoichiometry of 1:0.48 (± 0.005; *n* = 3) was determined after repeated simultaneous quantification of sulfide and oxygen” (Line 174). In addition, we have written down the stock concentrations that were added to the MIMS chamber in the methods section (line 498).

L175 – 189 This section does not seem to be described in methods, and perhaps a figure in Supplementary would help.

The microorganisms were cultured in a chemostat connected to an anoxic flask in which 100 mM Na₂S and 210 mM HCl were mixed (Supplementary Figure 1). Through a peristaltic pump, we were able to adjust the inflow of H₂S at any time. By gradually increasing the H₂S load throughout the day and by measuring the H₂S concentration in the gas outlet, we determined the maximum load that the cells could respire. We found that the cells were able to respire 156 μmol H₂S · min⁻¹ · g DW⁻¹, without the sulfide concentration in the outlet reaching concentration above 25 nmol · L⁻¹. The gas concentrations (in the gas inlet and outlet) were measured by injecting sample into a GC. We have now described this procedure of gradually increasing the sulfide load to determine the maximum H₂S conversion rate in more detailed in the methods section: “In order to determine the maximum H₂S conversion rate of the chemostat, the cells were gradually exposed to higher H₂S concentrations by regulating the peristaltic pump. The H₂S concentrations in the gas inlet and gas outlet were determined using gas chromatography (described in the subsection: Batch incubations and gas chromatography)”. We believe the explanation in the methods section together with the description of the results and reference to the rates in the table provide a good way to present this experiment without an additional figure in the Supplementary.

L193. “Increased up to 50%” – do the authors mean “increased by 50%”, otherwise please indicate 50% of what.

Thank you for your suggestion. We meant to say, “increased by”. We have modified it accordingly and stated that the total respiration increased by approx. 40% (closer to 40 than to 50%).

L 192 – 203 and 217 - 224. These sections are very confusing. Please indicate where these data are presented. I could not find any figure or table showing the cessation or reduction of H₂S consumption together with increase of respiration when methanol or hydrogen or formate was added. 20 – 40 μM sulfide does not seem to match with the concentrations given in Table 1 (10 – 20 and 40 – 80 μM).

Thank you for your suggestions. We have provided two figures (Figure 3a and 3b) that show cessation and reduction of H₂S consumption after addition of methanol to non-adapted and sulfide-adapted cells, respectively. Because in all MIMS experiments a fiber-optic oxygen sensor spot was used, we determined substrate conversion rates as well as respiration rates. To make it easy for the reader to read the text, we decided to primarily focus on stating the substrate conversion rates (mostly in Table 1 and in the figures), while in the main text we state the respiration rates as percentages (increase and decrease after substrate addition). To give the reader an idea of how we derived these respiration rates, we have prepared a graph (Supplementary Figure 4) in which we show H₂S respiration in the presence and absence of methanol. In addition, we prepared a graph (Supplementary Figure 5) in which we show the methanol respiration rates in the presence of various concentrations of H₂S.

The rates measured in Table 1 are from separate experiments, in which the maximum CH₄/H₂/methanol conversion and respiration states of the non-adapted and sulfide-adapted cells are stated in the absence of H₂S. The rationale of those experiments was studying the effect of growth in the presence of H₂S on metabolism. Because above an H₂S concentration of 30 μM only the sulfide-insensitive terminal oxidase seems to be active, we have performed multiple experiments looking at concentrations above and below 30 μM. We clearly see that respiration is affected by the H₂S concentration and that the sulfide-sensitive terminal oxidases partially participate in respiration at low H₂S concentrations, increasing the respiration rates (Table 1).

As you correctly state in the next comment, additional experiments were performed in which more than one substrate was present. The values obtained in these experiments are not stated in the table. For clarity we added the following to the Table title: All CH₄, CH₃OH and H₂ conversion and respiration rates measured in the MIMS chamber were determined in the absence of H₂S.

To make the experiments in which more than one substrate are followed easier to understand for the reader, we have prepared Figures 3 and 4, and Supplementary Figures 4, 5, 6 and 7). By visualizing the data, the reader can see when a certain compound was added, and its effect on metabolism. We also moved Table 1 up in the main text, to physically separate it from the experiments in which more than one compound is involved.

Finally, the following text was stated in the section in which experiments using multiple substrates are studied: “Due to the low dO_2 concentration in the continuous cultures, the non-adapted and sulfide-adapted cells expressed a high hydrogenase activity (Table 1), with a measured $H_2:O_2$ consumption ratio of approximately 1:0.35 as expected (32, 43). As was the case for CH_4 and methanol respiration, the maximum H_2 respiration rates of the sulfide-adapted cells were lower than those of the non-adapted cells (Table 1).” Because these rates were determined in the absence of H_2S , we have now moved this text up (Lines 135-139), and stated the rates in Table 1.

L201 “as opposed to methane” – where is this shown? My understanding is that Table 1 refers to cells removed from the two growth conditions and tested with the listed substrates in the MIMS chamber, whereas these text sections seem to refer to more complicated experiments in which more than one substrate was added sequentially. Unless I am misunderstanding completely, these sections refer to experiments which are not described or presented. Please clarify. I suggest a figure showing these changes in consumption, and referring to it in the text.

In Figure 2 we see how methane respiration in non-adapted cells (and the same is the case in sulfide-adapted cells) is affected by H_2S . For instance, in the presence of 10 μM , methane consumption is inhibited by 95%. As opposed to methane, methanol is still respired (Figure 3 and Supplementary Figure 4 and 5). Hence, the PMO seems to be the prime target of H_2S inhibition, rather than the methanol dehydrogenase.

You correctly understand that Table 1 refers to cells removed from the two growth conditions and were tested for their consumption of CH_4 , H_2 and methanol in the absence of H_2S , and H_2S consumption. We have now physically separated these sections by moving the Table up in the main text. Instead of stating “as opposed to methane”, we now state: “...indicating that at least some methanol was still respired, which was confirmed by the fact that the CO_2 production rate continued at 20–30% in the presence of sulfide (Supplementary Figure 5). In contrast, at the same sulfide concentrations, methane respiration had ceased almost completely (Figure 2).”

Table 1. Caption, “Respiration refers to O_2 consumption of the listed substrates” – should this be “in response to the listed substrates”?

Modified accordingly, thank you for the suggestion.

Table 1 What is meant by “ CH_4 respiration” under maximum rates? This figure seems to be arrived at by subtracting “ CH_4 conversion” from “ O_2 for CH_4 ”, and therefore, if I’m understanding correctly, is the expected O_2 consumption for the rest of the CH_4 consumption pathway, following the initial oxidation (?) in which case the term CH_4 respiration is confusing.

Indeed, we measured the maximum methane conversion rate ($200 \pm 11 \mu mol \cdot min^{-1} \cdot g DW^{-1}$) as well as the maximum O_2 consumption rate with methane as energy source ($302 \pm 9 \mu mol \cdot min^{-1} \cdot g DW^{-1}$). Because theoretically 1 mole O_2 is needed to activate 1 mole CH_4 , the maximum respiration rate on methane is $102 \pm 11 \mu mol \cdot min^{-1} \cdot g DW^{-1}$ (indeed, subtracting the rates). However, we understand this might be confusing. Instead, we have given the total respiration rate with methane as energy source ($302 \pm 9 \mu mol \cdot min^{-1} \cdot g DW^{-1}$) in the table, with a footnote indicating that theoretically 1 mole O_2 is needed to activate 1 mole CH_4 . We have also stated this in the results section (Lines 132-133).

L212 Consider prefacing “Due to” with “Likely”

Modified accordingly, thank you.

L222. Mostly the H₂S concentrations are given as a range, should this be 20 – 40 μM?

That is correct, thank you. We have modified it accordingly.

L226 – 228. I did not understand this sentence and I’m not sure how a terminal oxidase can have a preference for a substrate as such. Please consider if this sentence is appropriate and/or clarify.

Thank you for your suggestion. We would like to clarify our statement. We have measured that in the presence of sulfide, the respiration rate is limited. In the presence of high sulfide concentrations, a low but stable respiration rate is found, indicating the presence of a sulfide-insensitive terminal oxidase (SITO). Because of its limited capacity, we observed competition between methanol and sulfide, both in sulfide-adapted and non-adapted cells (Figure 3). We have thoroughly thought of how to explain this, and whether using “preference” for biochemical processes is perhaps too anthropomorphic. Instead, we now state “The observation that H₂S is only oxidized after H₂ (or methanol) has become depleted (Figure 4) suggests competitive electron transfer pathways to the sulfide-insensitive terminal oxidase (SITO) due to its limited respiration capacity.”

Hence, the point we make is that the electron transport chain from electrons derived from methanol oxidation is different than the electron transport chain from electrons derived from sulfide oxidation. For instance, in the case of methanol, electrons released through catalysis of the methanol dehydrogenase are transferred to the cytochrome *c* protein XoxGJ (Versantvoort et al., 2019). Hereafter, it is hypothesized that electrons are directly transferred to a terminal oxidase (Complex IV). In the case of H₂S oxidation, in contrast, the SQR is thought to donate electrons to the quinone pool, in which alternative complex III (ACIII) is involved. Different proteins involved in electron transport chains can explain why catalysis A is faster, or more favorable/preferred, than catalysis B. Very interestingly, the “preference” for electrons from methanol/H₂ versus H₂S is different in non-adapted and sulfide-adapted cells. As we show throughout the text, the cells adapt to sulfide by upregulating genes involved in the electron transport chain (encoding SQR, Complex IV and the putative heptahaem cytochrome *c* protein). Altogether, we now speak of “competitive electron transfer pathways” instead of “preference” of the terminal oxidase.

L237 – 238 If the authors are claiming that the difference here is due to high [O₂] and low [H₂S] please give the O₂ concentration and identify the rate in the MIMS chamber. From Table 1, unless I’m mistaken, this seems to be 100 +/- 11, so the rate is more than slightly higher. I don’t fully understand the phrase “partially alleviating inhibition” (L241) – inhibition of what and by what, and how does [O₂] play a role? Please rephrase and/or clarify.

Thank you for your comment. To clarify our point, we have added additional rates to the table and elaborated in the main text. In the table we state a value of $120 \pm 13 \mu\text{mol H}_2\text{S} \cdot \text{min}^{-1} \cdot \text{g DW}^{-1}$, which is the maximum H₂S conversion rate measured in the MIMS chamber of the sulfide adapted cells in the presence of 5–30 μM H₂S and < 10 μM O₂. In comparison, in batch incubations, maximum H₂S conversion rates of 167–223 μmol H₂S · min⁻¹ · g DW⁻¹ were measured using GC (starting at 3.5 μM H₂S and 190 μM O₂). You are correct that this is more than slightly higher. We forgot to mention in the table that we also measured H₂S consumption in the MIMS chamber in the presence of higher O₂ concentrations of 60 to 80 μM O₂. Under these conditions, we measured maximum H₂S conversion rates of 132 – 154 μmol H₂S · min⁻¹ · g DW⁻¹ by sulfide-adapted cells. Hence, what is measured at 60 to 80 μM O₂ is comparable (but slightly lower) than measured in batch incubations using GC. The small difference could be explained by the higher O₂ and low H₂S concentrations present in the incubations used for GC measurements. Because O₂ and H₂S

compete for the sulfide-sensitive terminal oxidase (SSTO), a low H₂S and high O₂ concentration partially alleviate SSTO inhibition because of a competition for the active site, leading to higher sulfide consumption rates. We have now clarified this in the text and added the rate measured 60 to 80 μM O₂ to the table.

L247 Should this be sequential addition of O₂?

Correct, thank you for noticing the mistake.

Fig 2a Please mention in the legend that the reactions were initiated by addition of cells and at which timepoints.

We have mentioned the timepoints at which the reactions were initiated, thank you.

L264 Since several mentions of SQR types are made throughout, it would be helpful to include, either in the intro or in the section commencing. L285, a sentence or two describing the basis of SQR type classification and what is known about the properties of the different groups. If space is limited, L42 – 45 could be omitted, since this passage about the harmful effects of H₂S is largely irrelevant in the context of this study.

The classification of SQR types is based on protein structures, as proposed by Marcia et al. in 2010. In addition, some SQR classes are more typical to certain groups of microorganisms than others (for instance, type IV SQRs are typical for green sulfur bacteria). The SQR classes also differ in their affinity for sulfide and their physiological role in the cell. We have added this information to the text.

We have omitted the part about harmful effects and now just state in which environments H₂S is produced through sulfate reduction, mineralization of organic matter and thermochemistry.

L274 H₂S or sulfide? Please give a few examples of the metabolites.

Indeed, genes involved in the production of sulfide were downregulated in the presence of sulfide. In the cell, sulfide is produced because it is needed for the synthesis of several sulfur-containing metabolites, such as cysteine, methionine and glutathione. We have added these examples to the manuscript.

L277 Delete “their”

We replaced “their” with “the measured”.

L278 conversation >> conversion

Modified accordingly, thank you.

Table 2 Why not use the same colors as Fig S7?

Thank you for the good suggestion. The colours in table 2 now correspond to the colours in Supplementary Figure 9.

L289 “different types” unnecessarily vague and emphasises the value of definition of SQR types

We have briefly explained in the text how Marci et al. (2010) classify the different SQR types. We would like to emphasize how widespread SQR is in a large variety found in many different environments. Because we hope that

our paper will enthruse readers to investigate SQRs and sulfide oxidation in methanotrophs, we have now elaborated on where the SQR-encoding methanotrophs are found, and briefly how SQRs are classified.

Fig 3 This tree is not very informative as is. I suggest identifying the characterised S-oxidizers. Define FCSD which has not yet been done. Specify amino acid or nucleotide and define scale bar.

Thank you for your suggestions. We have made a large phylogenetic tree (Supplementary Information 10), in which we have identified characterized S-oxidizers, defined FCSD, specified that we show amino acid sequences and defined the scale bar.

L311 delete “much more” or define more than what

We deleted “much more”.

L346 Please identify the highly upregulated gene (mfumv2_1950?)

We have added the gene identifier, thank you.

L351 – 355 Please supply a reference that these two polypeptides (XoxJG) function together as an electron carrier. Please indicate the extent of the change in ratio and clarify that this is gene transcripts, not protein (as suggested in the text).

We have provided the reference of Versantvoort et al. (2019; 10.1016/j.bbapap.2019.04.001), who have nicely shown that the XoxGJ fusion protein functions as the direct electron acceptor of the XoxF-type methanol dehydrogenase. The change in transcript ratio is quite large:

Mfum_v2_1185 (XoxGJ)

TPM: 524 + 91 (methane reactor)

TPM: 448 + 74 (methane-sulfide reactor)

Mfum_v2_1950 (putative multihæm electron carrier)

TPM: 19 + 2 (methane reactor)

TPM: 380 + 23 (methane-sulfide reactor)

We have indicated the extent of the change in ratio in the text and clarified that we talk about gene transcripts, not proteins: “In contrast, in non-adapted cells the ratio in transcripts of the genes encoding XoxGJ and the putative heptahæm electron carrier is 27.6 compared to 1.2 in sulfide-adapted cells. Accordingly, the upregulation the gene encoding the multihæm electron carrier might enable sulfide respiration to occur concurrently with methanol oxidation, using the same terminal oxidase.”

L360 – 362 please rephrase in more accurate and scientific language

We changed “*M. fumariolicum* SolV converts H₂S to elemental sulfur, but the reducing equivalents produced from this oxidation seem to muddle the electron transport chain.” into “Oxidation of H₂S is needed to keep this molecule at low, non-inhibitory concentrations. Consequently, the electrons released from this oxidation need to be processed by the electron transport chain, leading to substrate competition during the simultaneous oxidation of multiple compounds such as H₂S and CH₄.”

L376 High affinity constant = low affinity

Thank you for your suggestion. We meant to say, “a high affinity” and hence a low affinity constant. We have modified the text accordingly.

L421 – 2 and L482 If molarity is given, we don't need the water of hydration

We have modified the text accordingly.

L446 Please indicate the amount of cells in the chamber

Cells were taken from the continuous bioreactors and injected into the MIMS cells. The OD600 in the MIMS chamber ranged from approx. 0.02 to 0.1. We have not attempted to determine the number of cells that correspond to these optical densities, but we quantified that an optical density of 1 measured at 600 nm is equivalent to approximately 450 mg dry weight (DW) per litre. All the rates mentioned in the text, table and figures are stated as $\mu\text{mol} \cdot \text{min}^{-1} \cdot \text{g DW}^{-1}$ for easy comparison.

L450 shortly >> briefly

Modified in text, thank you.

L462 insert “measurement of” after “interference with” if that is what is meant

We have modified your suggestion in the manuscript, thank you.

L482 Define the calibration curve range. HCl added?

400 μL of a 25 mM Na_2S stock was acidified with 2 mL 0.5 M HCl in a 574 mL bottle creating a headspace concentration of 17.4 nmol/mL. Small volumes of the headspace were subsequently added to a 1162 mL bottle to create various H_2S concentrations to be injected (0.1 mL) into the GC for calibration. The calibration curve ranged from 1 pmol/mL to 1 nmol/mL. We have added this information to the method section.

In many places the tense is confusing. To report experimental results the past tense should be used. For example, L215, as was the case for CH_4 ... L307, cells responded to the presence...

We have modified the tenses accordingly, thank you.

Reviewer #1 (Remarks to the Author):

The authors answered all my questions and made appropriate comments regarding my concerns. I agree with the modified manuscript.

Reviewer #3 (Remarks to the Author):

I thank the authors for careful and comprehensive replies to my queries. This is a valuable study and I am happy to recommend publication. I have a few more or less trivial suggestions:

L113: cells became enriched

L223: H₂ consumption rate by 30%

L236: are used as electron >> was used

L273: MFUM_v2_0219-21 was upregulated

L278: 280, 286: was, were, was

L383: define ACIII

L396: encode sulfur

L487: operated at 40 uA

Reviewer #1 (Remarks to the Author):

The authors answered all my questions and made appropriate comments regarding my concerns. I agree with the modified manuscript.

Thank you very much for your previous suggestions.

Reviewer #3 (Remarks to the Author):

I thank the authors for careful and comprehensive replies to my queries. This is a valuable study and I am happy to recommend publication. I have a few more or less trivial suggestions:

Thank you very much for your compliments and for your excellent suggestions.

L113: cells became enriched

Modified accordingly, thank you.

L223: H2 consumption rate by 30%

Modified accordingly, thank you.

L236: are used as electron >> was used

Modified accordingly, thank you.

L273: MFUM_v2_0219-21 was upregulated

Modified accordingly, thank you.

L278: 280, 286: was, were, was

Modified accordingly, thank you.

L383: define ACIII

Defined accordingly, thank you.

L396: encode sulfur

Modified accordingly, thank you.

L487: operated at 40 uA

Modified accordingly, thank you.